# BRAIN NETWORK TRANSFORMER

**Xuan Kan**[1]  **Wei Dai**[2]  **Hejie Cui**[1]  **Zilong Zhang**[3]  **Ying Guo**[1]  **Carl Yang**[1]

[1]Emory University  [2]Stanford University  [3]University of International Business and Economics

{xuan.kan,hejie.cui,yguo2,j.carlyang}@emory.edu
dvd.ai@stanford.edu  201957020@uibe.edu.cn

## Abstract

Human brains are commonly modeled as networks of Regions of Interest (ROIs) and their connections for the understanding of brain functions and mental disorders. Recently, Transformer-based models have been studied over different types of data, including graphs, shown to bring performance gains widely. In this work, we study Transformer-based models for brain network analysis. Driven by the unique properties of data, we model brain networks as graphs with nodes of fixed size and order, which allows us to (1) use connection profiles as node features to provide natural and low-cost positional information and (2) learn pairwise connection strengths among ROIs with efficient attention weights across individuals that are predictive towards downstream analysis tasks. Moreover, we propose an ORTHONORMAL CLUSTERING READOUT operation based on self-supervised soft clustering and orthonormal projection. This design accounts for the underlying functional modules that determine similar behaviors among groups of ROIs, leading to distinguishable cluster-aware node embeddings and informative graph embeddings. Finally, we re-standardize the evaluation pipeline on the only one publicly available large-scale brain network dataset of ABIDE, to enable meaningful comparison of different models. Experiment results show clear improvements of our proposed BRAIN NETWORK TRANSFORMER on both the public ABIDE and our restricted ABCD datasets. The implementation is available at https://github.com/Wayfear/BrainNetworkTransformer.

## 1  Introduction

Brain network analysis has been an intriguing pursuit for neuroscientists to understand human brain organizations and predict clinical outcomes [50, 59, 58, 5, 18, 27, 52, 29, 58, 28, 41, 44, 31]. Among various neuroimaging modalities, functional Magnetic Resonance Imaging (fMRI) is one of the most commonly used for brain network construction, where the nodes are defined as Regions of Interest (ROIs) given an atlas, and the edges are calculated as pairwise correlations between the blood-oxygen-level-dependent (BOLD) signal series extracted from each region [54, 53, 59, 16]. Researchers observe that some regions can co-activate or co-deactivate simultaneously when performing cognitive-related tasks such as action, language, and vision. Based on this pattern, brain regions can be classified into diverse functional modules to analyze diseases towards their diagnosis, progress understanding and treatment.

Nowadays Transformer-based models have led a tremendous success in various downstream tasks across fields including natural language processing [56, 17] and computer vision [20, 10, 55]. Recent efforts have also emerged to apply Transformer-based designs to graph representation learning. GAT [57] firstly adapts the attention mechanism to graph neural networks (GNNs) but only considers the local structures of neighboring nodes. Graph Transformer [21] injects edge information into the attention mechanism and leverages the eigenvectors of each node as positional embeddings. SAN [40] further enhances the positional embeddings by considering both eigenvalues and eigenvectors and improves the attention mechanism by extending the attention from local to global structures.

36th Conference on Neural Information Processing Systems (NeurIPS 2022).

Graphomer [64], which achieves the first place on the quantum prediction track of OGB Large-Scale Challenge [30], designs unique mechanisms for molecule graphs such as centrality encoding to enhance node features and spatial/edge encoding to adapt attention scores.

However, brain networks have several unique traits that make directly applying existing graph Transformer models impractical. First, one of the simplest and most frequently used methods to construct a brain network in the neuroimaging community is via pairwise correlations between BOLD time courses from two ROIs [43, 35, 13, 63, 69]. This impedes the designs like centrality, spatial, and edge encoding because each node in the brain network has the same degree and connects to every other node by a single hop. Second, in previous graph transformer models, eigenvalues and eigenvectors are commonly used as positional embeddings because they can provide identity and positional information for each node [15, 26]. Nevertheless, in brain networks, the connection profile, which is defined as each node's corresponding row in the brain network adjacency matrix, is recognized as the most effective node feature [13]. This node feature naturally encodes both structural and positional information, making the aforementioned positional embedding design based on eigenvalues and eigenvectors redundant. The third challenge is scalability. Typically, the numbers of nodes and edges in molecule graphs are less than 50 and 2500, respectively. However, for brain networks, the node number is generally around 100 to 400, while the edge number can be up to 160,000. Therefore, operations like the generation of all edge features in existing graph transformer models can be time-consuming, if not infeasible.

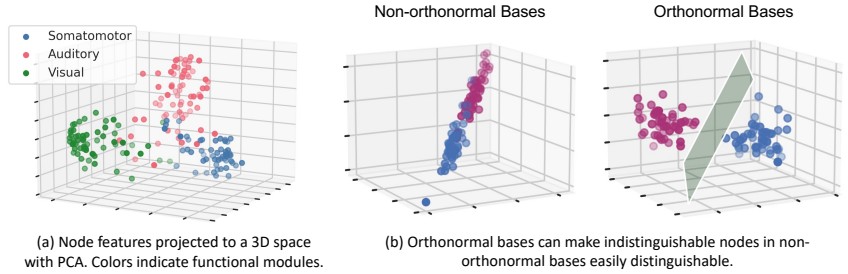

(a) Node features projected to a 3D space with PCA. Colors indicate functional modules.

(b) Orthonormal bases can make indistinguishable nodes in non-orthonormal bases easily distinguishable.

Figure 1: Illustration of the motivations behind ORTHONORMAL CLUSTERING READOUT.

In this work, we propose to develop BRAIN NETWORK TRANSFORMER (BRAINNETTF), which leverages the unique properties of brain network data to fully unleash the power of Transformer-based models for brain network analysis. Specifically, motivated by previous findings on effective GNN designs for brain networks [13], we propose to use the effective initial node features of connection profiles. Empirical analysis shows that connection profiles naturally provide positional features for Transformer-based models and avoid the costly computations of eigenvalues or eigenvectors. Moreover, recent work demonstrates that GNNs trained on learnable graph structures can achieve superior effectiveness and explainability [35]. Inspired by this insight, we propose to learn fully pairwise attention weights with Transformer-based models, which resembles the process of learning predictive brain network structures towards downstream tasks.

One step further, when GNNs are used for brain network analysis, a graph-level embedding needs to be generated through a readout function based on the learned node embeddings [37, 43, 13]. As is shown in Figure 1(a), a property of brain networks is that brain regions (nodes) belonging to the same functional modules often share similar behaviors regarding activations and deactivations in response to various stimulations [7]. Unfortunately, the current labeling of functional modules is rather empirical and far from accurate. For example, [3] provides more than 100 different functional module organizations based on hierarchical clustering. In order to leverage the natural functions of brain regions without the limitation of inaccurate functional module labels, we design a new global pooling operator, ORTHONORMAL CLUSTERING READOUT, where the graph-level embeddings are pooled from clusters of functionally similar nodes through soft clustering with orthonormal projection. Specifically, we first devise a self-supervised mechanism based on [60] to jointly assign soft clusters to brain regions while learning their individual embeddings. To further facilitate the learning of clusters and embeddings, we design an orthonormal projection and theoretically prove its effectiveness in distinguishing embeddings across clusters, thus obtaining expressive graph-level embeddings after the global pooling, as illustrated in Figure 1(b).

Finally, the lack of open-access datasets has been a non-negligible challenge for brain network analysis. The strict access restrictions and complicated extraction/preprocessing of brain networks from fMRI data limit the development of machine learning models for brain network analysis. Specifically, among all the large-scale publicly available fMRI datasets in literature, ABIDE [6] is the only one provided with extracted brain networks fully accessible without permission requirements. However, ABIDE is aggregated from 17 international sites with different scanners and acquisition parameters. This inter-site variability conceals inter-group differences that are really meaningful, which is reflected in the unstable training performance and the significant gap between validation and testing performance in practice. To address these limitations, we propose to apply a stratified sampling method in the dataset splitting process and standardize a fair evaluation pipeline for meaningful model comparison on the ABIDE dataset. Our extensive experiments on this public ABIDE dataset and a restricted ABCD dataset [8] show significant improvements brought by our proposed BRAIN NETWORK TRANSFORMER.

## 2 Background and Related Work

### 2.1 GNNs for Brain Network Analysis

Recently, emerging attention has been devoted to the generalization of GNN-based models to brain network analysis [42, 2]. GroupINN [62] utilizes a grouping-based layer to provide explainability and reduce the model size. BrainGNN [43] designs the ROI-aware GNNs to leverage the functional information in brain networks and uses a special pooling operator to select these crucial nodes. IBGNN [14] proposes an interpretable framework to analyze disorder-specific ROIs and prominent connections. In addition, FBNetGen [35] considers the learnable generation of brain networks and explores the explainability of the generated brain networks towards downstream tasks. Another benchmark paper [13] systematically studies the effectiveness of various GNN designs over brain network data. Different from other work focusing on static brain networks, STAGIN [39] utilizes GNNs with spatio-temporal attention to model dynamic brain networks extracted from fMRI data.

### 2.2 Graph Transformer

Graph Transformer raises many researchers' interest currently due to its outstanding performance in graph representation learning. Graph Transformer [21] firstly injects edge information into the attention mechanism and leverages the eigenvectors as positional embeddings. SAN [40] enhances the positional embeddings and improves the attention mechanism by emphasizing neighbor nodes while incorporating the global information. Graphomer [64] designs unique mechanisms for molecule graphs and achieves the SOTA performance. Besides, a fine-grained attention mechanism is developed for node classification [68]. Also, the Transformer is extended to larger-scale heterogeneous graphs with a particular sampling algorithm in HGT [32]. EGT [33] further employs edge augmentation to assist global self-attention. In addition, LSPE [22] leverages the learnable structural and positional encoding to improve GNNs' representation power, and GRPE [49] enhances the design of encoding node relative position information in Transformer.

## 3 BRAIN NETWORK TRANSFORMER

### 3.1 Problem Definition

In brain network analysis, given a brain network $\boldsymbol{X} \in \mathbb{R}^{V \times V}$, where $V$ is the number of nodes (ROIs), the model aims to make a prediction indicating biological sex, presence of a disease or other properties of the brain subject. The overall framework of our proposed BRAIN NETWORK TRANSFORMER is shown in Figure 2, which is mainly composed of two components, an $L$-layer attention module MHSA and a graph pooling operator OCREAD. Specifically, in the first component of MHSA, the model learns attention-enhanced node features $\boldsymbol{Z}^L$ through a non-linear mapping $\boldsymbol{X} \rightarrow \boldsymbol{Z}^L \in \mathbb{R}^{V \times V}$. Then the second component of OCREAD compresses the enhanced node embeddings $\boldsymbol{Z}^L$ to graph-level embeddings $\boldsymbol{Z}_G \in \mathbb{R}^{K \times V}$, where $K$ is a hyperparameter representing the number of clusters. $\boldsymbol{Z}_G$ is then flattened and passed to a multi-layer perceptron for graph-level predictions. The whole training process is supervised with the cross-entropy loss.

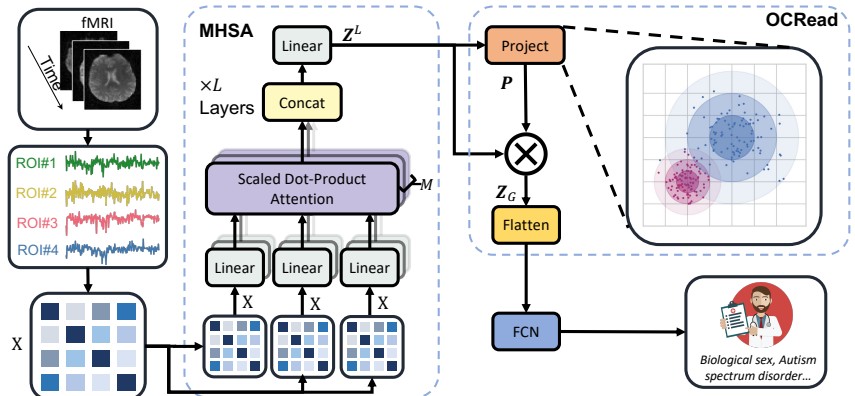

Figure 2: The overall framework of our proposed BRAIN NETWORK TRANSFORMER.

## 3.2 Multi-Head Self-Attention Module (MHSA)

To develop a powerful Transformer-based model suitable for brain networks, two fundamental designs, the positional embedding and attention mechanism, need to be reconsidered to fit the natural properties of brain network data. In existing graph transformer models, the positional information is usually encoded via eigendecomposition, while the attention mechanism often combines node positions with existing edges to calculate the attention scores. However, for the dense (often fully connected) graphs of brain networks, eigendecomposition is rather costly, and the existence of edges is hardly informative.

ROI node features on brain networks naturally contain sufficient positional information, making the positional embeddings based on eigendecomposition redundant. Previous work on brain network analysis has shown that the connection profile $\boldsymbol{X}_{i\cdot}$ for node $i$, defined as the corresponding row for each node in the edge weight matrix $\boldsymbol{X}$, always achieves superior performance over others such as node identities, degrees or eigenvector-based embeddings [43, 35, 13]. With this node feature initialization, the self-connection weight $\boldsymbol{x}_{ii}$ on the diagonal is always equal to one, which encodes sufficient information to determine the position of each node in a fully connected graph based on the given brain atlas. To verify this insight, we also empirically compare the performance of the original connection profile with two variants concatenated with additional positional information, *i.e.*, connection profile w/ identity feature and connection profile w/ eigen feature. The results indeed show no benefit brought by the additional computations (c.f. Appendix B). As for the attention mechanism, previous work [13] has empirically demonstrated that integrating edge weights into the attention score calculation can significantly degrade the effectiveness of attention on complete graphs, while the generation of edge-wise embedding can be unaffordable given a large number of edges in brain networks. On the other hand, the existence of edges provides no useful information for the computation of attention scores as well because all edges simply exist in complete graphs.

Based on the observations above, we design the basic BRAIN NETWORK TRANSFORMER by (1) adopting the connection profile as initial node features and eliminating any extra positional embeddings and (2) adopting the vanilla pair-wise attention mechanism without using edge weights or relative position information to learn a singular attention score for each edge in the complete graph.

Formally, we leverage a $L$-layer non-linear mapping module, namely Multi-Head Self-Attention (MHSA), to generate more expressive node features $\boldsymbol{Z}^L = \mathrm{MHSA}(\boldsymbol{X}) \in \mathbb{R}^{V \times V}$. For each layer $l$, the output $\boldsymbol{Z}^l$ is obtained by

$$\boldsymbol{Z}^l = (\|_{m=1}^{M} \boldsymbol{h}^{l,m})\boldsymbol{W}_{\mathcal{O}}^l, \boldsymbol{h}^{l,m} = \mathrm{Softmax}\left(\frac{\boldsymbol{W}_{\mathcal{Q}}^{l,m}\boldsymbol{Z}^{l-1}(\boldsymbol{W}_{\mathcal{K}}^{l,m}\boldsymbol{Z}^{l-1})^{\top}}{\sqrt{d_{\mathcal{K}}^{l,m}}}\right)\boldsymbol{W}_{\mathcal{V}}^{l,m}\boldsymbol{Z}^{l-1}, \quad (1)$$

where $\boldsymbol{Z}^0 = \boldsymbol{X}$, $\|$ is the concatenation operator, $M$ is the number of heads, $l$ is the layer index, $\boldsymbol{W}_{\mathcal{O}}^l, \boldsymbol{W}_{\mathcal{Q}}^{l,m}, \boldsymbol{W}_{\mathcal{K}}^{l,m}, \boldsymbol{W}_{\mathcal{V}}^{l,m}$ are learnable model parameters, and $d_{\mathcal{K}}^{l,m}$ is the first dimension of $\boldsymbol{W}_{\mathcal{K}}^{l,m}$.

### 3.3 ORTHONORMAL CLUSTERING READOUT (OCREAD)

The readout function is an essential component to learn the graph-level representations for brain network analysis (*e.g.*, classification), which maps a set of learned node-level embeddings to a graph-level embedding. $\mathrm{Mean}(\cdot), \mathrm{Sum}(\cdot)$ and $\mathrm{Max}(\cdot)$ are the most commonly used readout functions for GNNs. Xu et al. [61] show that GNNs equipped with $\mathrm{Sum}(\cdot)$ readout have the same discriminative power as the Weisfeiler-Lehman Test. Zhang et al. [66] propose a sort pooling to generate the graph-level representation by sorting the final node representations. Ju et al. [34] present a layer-wise readout by extending the node information aggregated from the last layer of GNNs to all layers. However, none of the existing readout functions leverages the properties of brain networks that nodes in the same functional modules tend to have similar behaviors and clustered representations, as shown in Figure 1(a). To address this deficiency, we design a novel readout function to take advantage of the modular-level similarities between ROIs in brain networks, where nodes are assigned softly to well-chosen clusters with an unsupervised process.

Formally, given $K$ cluster centers, each center has $V$ dimensions, $\boldsymbol{E} \in \mathbb{R}^{K \times V}$, a Softmax projection operator is used as the function to calculate the probability $\boldsymbol{P}_{ik}$ of assigning node $i$ to cluster $k$,

$$\boldsymbol{P}_{ik} = \frac{e^{\langle \boldsymbol{Z}_{i\cdot}^{L}, \boldsymbol{E}_{k\cdot} \rangle}}{\sum_{k'}^{K} e^{\langle \boldsymbol{Z}_{i\cdot}^{L}, \boldsymbol{E}_{k'\cdot} \rangle}}, \tag{2}$$

where $\langle \cdot, \cdot \rangle$ denotes the inner product and $\boldsymbol{Z}^{L}$ is the learned set of node embeddings from the last layer of MHSA module. With this computed soft assignment $\boldsymbol{P} \in \mathbb{R}^{V \times K}$, the original learned node representation $\boldsymbol{Z}^{L}$ can be aggregated under the guidance of the soft cluster information, where the graph-level embedding $\boldsymbol{Z}_G$ is obtained by $\boldsymbol{Z}_G = \boldsymbol{P}^{\top} \boldsymbol{Z}^{L}$.

However, jointly learning node embeddings and clusters without ground-truth cluster labels is difficult. To obtain representative soft assignment $\boldsymbol{P}$, the initialization of $K$ cluster centers $\boldsymbol{E}$ is critical and should be designed delicately. To this end, we leverage the observation illustrated in Figure 1(b), where orthonormal embeddings can improve the clustering of nodes in brain networks *w.r.t.* the functional modules underlying brain regions.

**Orthonormal Initialization**. To initialize a group of orthonormal bases as cluster centers, we first adopt the Xavier uniform initialization [25] to initialize $K$ random centers and each center contains $V$ dimensions $\boldsymbol{C} \in \mathbb{R}^{K \times V}$. Then, we apply the Gram-Schmidt process to obtain the orthonormal bases $\boldsymbol{E}$, where

$$\boldsymbol{u}_k = \boldsymbol{C}_{k\cdot} - \sum_{j=1}^{k-1} \frac{\langle \boldsymbol{u}_j, \boldsymbol{C}_{k\cdot} \rangle}{\langle \boldsymbol{u}_j, \boldsymbol{u}_j \rangle} \boldsymbol{u}_j, \quad \boldsymbol{E}_{k\cdot} = \frac{\boldsymbol{u}_k}{\|\boldsymbol{u}_k\|}. \tag{3}$$

In the next section, we theoretically prove the advantage of this orthonormal initialization.

#### 3.3.1 Theoretical Justifications

In OCREAD, proper cluster centers can generate higher-quality soft assignments and enlarge the difference between $\boldsymbol{P}$ from different classes. [51, 46] showed the advantages of orthogonal initialization in DNN model parameters. However, none of them proves whether it is an ideal strategy to obtain the cluster centers. We propose two methods from the perspective of statistics as follows.

Firstly, to discern features of different nodes, we would expect a larger discrepancy among their similarity probabilities indicated from the readout. One way to measure the discrepancy is using the *variance* of $\boldsymbol{P}$ for each feature. Let $\bar{\boldsymbol{P}} \equiv 1/K$ denote the mean of any discrete probabilities with $K$ values. Variance of $\boldsymbol{P}$ measures the difference between $\boldsymbol{P}$ and $\bar{\boldsymbol{P}}$. We average over the feature vector space: if the result is small, then there is a large tendency that different $\boldsymbol{P}$ approaches $\bar{\boldsymbol{P}}$ and hence cannot be discerned easily. Specifically, the following theorem holds for our function Eq. (2):

**Theorem 3.1.** *For arbitrary $r > 0$, let $B_r = \{\mathcal{Z} \in \mathbb{R}^V ; \|\mathcal{Z}\| \le r\}$ denote the round ball centered at origin of radius $r$ with $\mathcal{Z}$ being fracture vectors. Let $V_r$ be the volume of $B_r$. The variance of Softmax projection averaged over $B_r$*

$$\frac{1}{V_r} \int_{B_r} \sum_{k}^{K} \left( \frac{e^{\langle \mathcal{Z}, \boldsymbol{E}_{k\cdot} \rangle}}{\sum_{k'}^{K} e^{\langle \mathcal{Z}, \boldsymbol{E}_{k'\cdot} \rangle}} - \frac{1}{K} \right)^2 d\mathcal{Z}, \tag{4}$$

*attains maximum when $\boldsymbol{E}$ is orthonormal.*

Despite the concise form, it is unclear whether the above integral has an elementary antiderivative. Even though, we can circumvent this problem and a rigorous proof is given in Appendix C.

The second statistical method shows that for general readout functions without a known analytical form, initializing with orthonormal cluster centers has a larger probability of gaining better performance. To set up the proper statistical scenario, we assume that the unknown readout is obtained by a regression of some samples $(\hat{\boldsymbol{Z}}^{(s)}, \hat{\boldsymbol{E}}^{(t)}, \hat{\boldsymbol{P}}^{(st)})$. This formally converts the exact functional relationship between $\boldsymbol{Z}_{i\cdot}, \boldsymbol{E}_{k\cdot}$ and $\boldsymbol{P}_{ik}$ to a *statistical relationship*:

$$\boldsymbol{P}_T(\boldsymbol{Z}_{i\cdot}, \boldsymbol{E}_{k\cdot}) = \boldsymbol{P}(\boldsymbol{Z}_{i\cdot}, \boldsymbol{E}_{k\cdot}) + \epsilon_i, \quad \epsilon_i \sim N(0, \sigma^2), \quad E(\epsilon_i) = 0, \quad D(\epsilon_i) = \sigma^2, \qquad (5)$$

with $\boldsymbol{P}_T$ being the probability *truly* reflecting similarities between nodes and clusters and $\epsilon_i$ denoting the stochastic error. It is almost impossible to find $\boldsymbol{P}_T$, but by computing the so-called *variation inflation factor* [47], we show that regression in orthonormal case has a higher accuracy than that in non-orthonormal case. Combining with a hypothesis testing, we obtain the following

**Theorem 3.2.** *The significance level $\alpha_{E_{k\cdot}}$ which reveals the probability of rejecting a well-estimated pooling is lower when sampling from orthonormal centers than that from non-orthonormal centers.*

More details can be seen in Appendix C.

### 3.4 Generalizing OCREAD to Other Graph Tasks and Domains

In this work, we tested the proposed OCREAD on functional connectivity (FC) based brain networks. Other popular modalities of brain networks include structural connectivities (SC), which describe the anatomical organization of the brain by measuring the fiber tracts between brain regions [4]. In SC-based brain networks, ROIs that are positionally close to each other on the structural connectivity networks tend to share similar connection profiles. This means the idea of OCREAD is also naturally applicable to SC networks, where the orthonormal clustering is based on the physical distances instead of the functional modules on FC.

At a higher level, the idea of our proposed OCREAD is not confined to graph-level prediction tasks on brain networks but can also be generalized to other graph learning tasks and domains. Precisely, there is a growing tendency in node/edge level prediction tasks to enhance the node/edge representation learning by utilizing the subgraph embeddings around each target node/edge [67, 65]. In this process, substructure learning needs to be performed on the subgraphs, where our proposed OCREAD can be adapted for compressing a set of node embeddings to subgraph embeddings. Besides, OCREAD is also potentially useful for other types of graphs in the biomedical domains. For example, for protein-protein interaction networks, proteins can be implicitly grouped by families that share common evolutionary origins [48], whereas for gene expression networks, genes can be grouped based on the latent pathway information [36]. Both of them are potential directions for the future application of OCREAD, among many others driven by biological or other types of prior knowledge regarding underlying node/edge groups.

## 4 Experiments

This section evaluates the effectiveness of our proposed BRAIN NETWORK TRANSFORMER (BRAINNETTF) with extensive experiments. We aim to address the following research questions:

**RQ1.** How does BRAINNETTF perform compared with state-of-the-art models of various types?

**RQ2.** How does our proposed OCREAD module perform with different model choices?

**RQ3.** Does the learned model of BRAINNETTF exhibit consistency with existing neuroscience knowledge and suggest reasonable explainability?

### 4.1 Experimental Settings

**Datasets.** We conduct experiments on two real-world fMRI datasets. (a) *Autism Brain Imaging Data Exchange (ABIDE)*: This dataset collects resting-state functional magnetic resonance imaging (rs-fMRI) data from 17 international sites, and all data are anonymous [6]. The used dataset contains brain networks from 1009 subjects, with 516 (51.14%) being Autism spectrum disorder

(ASD) patients (positives). The region definition is based on Craddock 200 atlas [12]. As the most convenient open-source large-scale dataset, it provides generated brain networks and can be downloaded directly without permission request. Despite the ease of acquisition, the heterogeneity of the data collection process hinders its use. Since multi-site data are collected from different scanners with different acquisition parameters, non-neural inter-site variability may mask inter-group differences. In practice, we find the training unstable, and there is a significant gap between validation and testing performances. However, we discover that most models can achieve a stable performance if we follow an appropriate stratified sampling strategy by considering collection sites during the training-validation-testing splitting process for ABIDE. Training curves in Appendix A also show how different models achieve a stabler performance on our designed new splitting settings than the random splitting. Therefore, we use ABIDE as one of the benchmark datasets in this work, and we share our re-standardized data splitting to provide a fair evaluation pipeline for various future methods. (b) *Adolescent Brain Cognitive Development Study (ABCD)*: This is one of the largest publicly available fMRI datasets with restricted access (a strict data requesting process needs to be followed to obtain the data) [8]. The data we use in the experiments are fully anonymized brain networks with only biological sex labels. After the quality control process, 7901 subjects are included in the analysis, with 3961 (50.1%) among them being female. The region definition is based on the HCP 360 ROI atlas [24].

**Metrics.** The diagnosis of ASD is the prediction target on ABIDE, while biological sex prediction is used as the evaluation task for ABCD. Both prediction tasks are binary classification problems, and both datasets are balanced between classes. Hence, AUROC is a proper performance metric adopted for fair comparison at various threshold settings, and accuracy is applied to reflect the prediction performance when the threshold is 0.5. Besides, since the model is mainly for medical applications, we add two critical metrics for diagnostic tests, Sensitivity and Specificity, which respectively refer to true positive rate and true negative rate. All reported performances are the average of 5 random runs on the test set with the standard deviation.

**Implementation details.** For experiments, we use a two-layer Multi-Head Self-Attention Module and set the number of heads $M$ to 4 for each layer. We randomly split 70% of the datasets for training, 10% for validation, and the remaining are utilized as the test set. In the training process of BRAINNETTF, we use an Adam optimizer with an initial learning rate of $10^{-4}$ and a weight decay of $10^{-4}$. The batch size is set as 64. All models are trained for 200 epochs, and the epoch with the highest AUROC performance on the validation set is used for performance comparison on the test set. The model is trained on an NVIDIA Quadro RTX 8000. Please refer to the repository and Appendix G for the full implementation of BRAINNETTF.

**Computation complexity.** In BRAINNETTF, the computation complexity of Multi-Head Self-Attention Module and OCREAD are $\mathcal{O}(LMV^2)$ and $\mathcal{O}(KV)$ respectively, where $L$ is the layer number of Multi-Head Self-Attention Module, $V$ is the number of nodes, $M$ is the number of heads, and $K$ is the number of clusters in OCREAD. The overall computation complexity of BRAINNETTF is thus $\mathcal{O}(V^2)$, which is on the same scale as common GNNs on brain networks such as BrainGNN [43] and BrainGB [13].

## 4.2 Performance Analysis (RQ1)

We compare BRAINNETTF with baselines of three types. The details about how to tune hyperparameters of various baselines can be found in Appendix F. Besides, Appendix E shows the comparison of the number of parameters between our model and other baseline models, which shows that the parameter size of BRAINNETTF is larger than GNN and CNN models but smaller than other transformer models. **(a) BRAINNETTF vs. other graph transformers.** We compare BRAINNETTF with two popular graph Transformers, SAN [40] and Graphormer [64]. In addition, we also include a basic version of BRAINNETTF without OCREAD, composed of a Transformer with a 2-layer Multi-Head Self-Attention and a CONCAT-based readout named VanillaTF. Our BRAINNETTF outperforms SAN and Graphormer by significant margins, with up to 6% absolute improvements on both datasets. VanillaTF also surpasses SAN and Graphormer. We believe this downgraded performance of existing graph transformers results from their design flaws facing the natures of brain networks. Specifically, both the preprocessing and the training stages of the Graphormer model accepts only discrete, categorical data. A bin operator has to be applied on the adjacency matrix, coarsening the node feature from connection profiles and dramatically hurting the performance.

Furthermore, since brain networks are complete graphs, key designs like centrality encoding and spatial encoding of Graphormer cannot be appropriately applied. Similarly, for SAN, experiments in Appendix B show that adding eigen node features to connection profiles cannot improve the model's performance. Besides, the benchmark paper [13] reveals that injecting edge weights into the attention mechanism can significantly reduce the prediction power. Furthermore, Appendix D shows our BRAINNETTF is much faster than other graph transformers due to special optimizations towards brain networks. **(b) BRAINNETTF vs. neural network models on fixed brain networks.** We further introduce another three neural network baselines on fixed brain networks. BrainGNN [43] designs ROI-aware GNNs for brain network analysis. BrainGB [13] is a systematic study of how to design effective GNNs for brain network analysis. We adopt their best design as the BrainGB baseline. BrainnetCNN [37] represents state-of-the-art of specialized GNNs for brain network analysis, which models the adjacency matrix of a brain network similarly as a 2D image. As is shown in Table 1, BRAINNETTF consistently outperforms BrainGNN, BrainGB and BrainnetCNN. **(c) BRAINNETTF vs. neural network models on learnable brain networks.** Unlike classical GNNs, FBNETGEN [35], DGM [38] and BrainNetGNN [45] hold a similar idea, which is to apply GNNs based on a learnable graph. FBNETGEN achieves SOTA performance on the ABCD dataset for biological sex prediction, and the learnable graphs can be seen as a type of attention score. Experiment results show that our proposed BRAINNETTF beats all three of them on both datasets.

Table 1: Performance comparison with different baselines (%). The performance gains of BRAIN-NETTF over the baselines have passed the t-test with p-value$<$0.03.

| Type | Method | Dataset: ABIDE | | | | Dataset: ABCD | | | |
|---|---|---|---|---|---|---|---|---|---|
| | | AUROC | Accuracy | Sensitivity | Specificity | AUROC | Accuracy | Sensitivity | Specificity |
| Graph Transformer | SAN | 71.3±2.1 | 65.3±2.9 | 55.4±9.2 | 68.3±7.5 | 90.1±1.2 | 81.0±1.3 | 84.9±3.5 | 77.5±4.1 |
| | Graphormer | 63.5±3.7 | 60.8±2.7 | **78.7±22.3** | 36.7±23.5 | 89.0±1.4 | 80.2±1.3 | 81.8±11.6 | 82.4±7.4 |
| | VanillaTF | 76.4±1.2 | 65.2±1.2 | 66.4±11.4 | 71.1±12.0 | 94.3±0.7 | 85.9±1.4 | 87.7±2.4 | 82.6±3.9 |
| Fixed Network | BrainGNN | 62.4±3.5 | 59.4±2.3 | 36.7±24.0 | 70.7±19.3 | OOM | OOM | OOM | OOM |
| | BrainGB | 69.7±3.3 | 63.6±1.9 | 63.7±8.3 | 60.4±10.1 | 91.9±0.3 | 83.1±0.5 | 84.6±4.3 | 81.5±3.9 |
| | BrainNetCNN | 74.9±2.4 | 67.8±2.7 | 63.8±9.7 | 71.0±10.2 | 93.5±0.3 | 85.7±0.8 | 87.9±3.4 | 83.0±4.4 |
| Learnable Network | FBNETGNN | 75.6±1.2 | 68.0±1.4 | 64.7±8.7 | 62.4±9.2 | 94.5±0.7 | 87.2±1.2 | 87.0±2.5 | 86.7±2.8 |
| | BrainNetGNN | 55.3±1.9 | 51.2±5.4 | 67.7±37.5 | 33.9±34.2 | 75.3±5.2 | 67.5±4.7 | 67.7±5.7 | 68.0±6.5 |
| | DGM | 52.7±3.8 | 60.7±12.6 | 53.8±41.2 | 51.1±40.9 | 76.8±19.0 | 68.6±8.1 | 40.5±29.7 | **95.6±4.2** |
| Ours | BRAINNETTF | **80.2±1.0** | **71.0±1.2** | 72.5±5.2 | **69.3±6.5** | **96.2±0.3** | **88.4±0.4** | **89.4±2.6** | 88.4±1.5 |

## 4.3 Ablation Studies on the OCREAD Module (RQ2)

### 4.3.1 OCREAD with varying readout functions

We vary the readout function for various Transformer architectures, including SAN, Graphormer and VanillaTF, to observe the performance of each ablated model variant. The results shown in Table 2 demonstrate that our OCREAD is the most effective readout function for brain networks and improves the prediction power across various Transformer architectures.

Table 2: Performance comparison AUROC (%) with different readout functions.

| Readout | Dataset: ABIDE | | | Dataset: ABCD | | |
|---|---|---|---|---|---|---|
| | SAN | Graphormer | VanillaTF | SAN | Graphormer | VanillaTF |
| MEAN | 63.7±2.4 | 50.1±1.1 | 73.4±1.4 | 88.5±0.9 | 87.6±1.3 | 91.3±0.7 |
| MAX | 61.9±2.5 | 54.5±3.6 | 75.6±1.4 | 87.4±1.1 | 81.6±0.8 | 94.4±0.6 |
| SUM | 62.0±2.3 | 54.1±1.3 | 70.3±1.6 | 84.2±0.8 | 71.5±0.9 | 91.6±0.6 |
| SortPooling | 68.7±2.3 | 51.3±2.2 | 72.4±1.3 | 84.6±1.1 | 86.7±1.0 | 89.9±0.6 |
| DiffPool | 57.4±5.2 | 50.5±4.7 | 62.9±7.3 | 78.1±1.5 | 70.0±1.9 | 83.9±1.3 |
| CONCAT | **71.3±2.1** | 63.5±3.7 | 76.4±1.2 | 90.1±1.2 | 89.0±1.4 | 94.3±0.7 |
| OCREAD | 70.6±2.4 | **64.9±2.7** | **80.2±1.0** | **91.2±0.7** | **90.2±0.7** | **96.2±0.4** |

### 4.3.2 OCREAD with varying cluster initializations

To further demonstrate how the design of OCREAD influences the performance of BRAINNETTF, we investigate two key model selections, the initialization method for cluster centers and the cluster

number $K$. For the initialization, three different kinds of initialization procedures are compared, namely (a) **Random**: the Xavier uniform [25] is leveraged to randomly generate a group of centers, which are then normalized into unit vectors; (b) **Learnable**: the same initial process as Random, but the generated centers are further updated with gradient descent; (c) **Orthonormal**: our proposed process as described in Eq. (3).

Specifically, we test each initialization method with the cluster number $K$ equals to 2, 3, 4, 5, 10, 50, 100. The results of adjusting these two hyper-parameters on ABIDE and ABCD datasets are shown in Figure 3(a). We observe that: (1) When cluster centers are orthonormal, the model's performance increases with the number of clusters ranging from 2 to 10, and then drops with the cluster number rising from 10 to 100, suggesting the optimal cluster number to be relatively small, which leads to less computation and is consistent with the fact that the typical number of functional modules are smaller than 25; (2) With a sufficiently large cluster number, all three initialization methods, Random, Learnable and Orthonormal, tend to reach similar performance, but orthonormal performs stably better when the number of clusters is smaller; (3) It is also notable that our OCREAD consistently achieves the best performance over other initialization methods regarding smaller standard deviations.

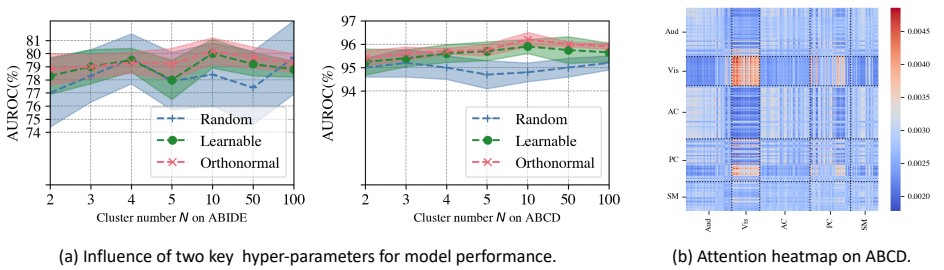

(a) Influence of two key hyper-parameters for model performance.          (b) Attention heatmap on ABCD.

Figure 3: The hyper-parameter influence and the heatmap from self-attention.

## 4.4 In-depth Analysis of Attention Scores and Cluster Assignments (RQ3)

Figure 3(b) displays the self-attention score from the first layer of Multi-Head Self-Attention. The attention scores are the average across all subjects in the ABCD test set. This figure shows that the learned attention scores well match the divisions of functional modules based on available labels, demonstrating the effectiveness and explainability of our Transformer model. Note that since there exists no available functional module labels for the atlas of the ABIDE dataset, we cannot visualize the correlations between attention scores and functional modules.

Figure 4 shows the cluster soft assignment results $P$ on nodes in OCREAD with two initialization methods. The cluster number $K$ is set to 4. The visualized numerical values are the average $P$ of all subjects in each dataset's test set. From the visualization, we observe that (a) Base on Appendix H, orthonormal initialization produces more discriminative $P$ between classes than random initialization; (b) Within each class, orthonormal initialization encourages the nodes to form groups. These observations demonstrate that our OCREAD with orthonormal initialization can leverage potential clusters underlying node embeddings, thus automatically grouping brain regions into potential functional modules.

## 5 Discussion and Conclusion

Neuroimaging technologies, including functional magnetic resonance imaging (fMRI) are powerful noninvasive tools for examining the brain functioning. There is an emerging nation-wide interest in conducting neuroimaging studies for investigating the connection between the biology of the brain, and demographic variables and clinical outcomes such as mental disorders. Such studies provide an unprecedented opportunity for cross-cutting investigations that may offer new insights to the differences in brain function and organization across subpopulations in the society (such as biological sex and age groups) as well as reveal neurophysiological mechanisms underlying brain disorders (such as psychiatric illnesses and neurodegenerative diseases). These studies have a tremendous impact in social studies and biomedical sciences. For example, mental disorders are the leading cause of disability in the USA and roughly 1 in 17 have a seriously debilitating mental illness. To

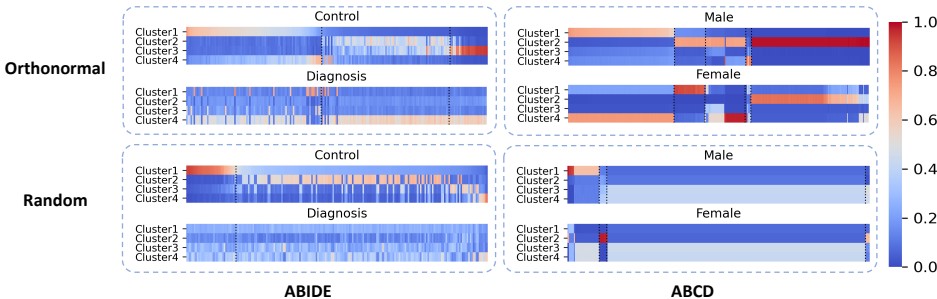

Figure 4: Visualization of cluster (module-level) embeddings learned with Orthonormal vs. Random cluster center initializations on two datasets. Each group in the dotted box contains two heatmaps (one for each prediction class) with the same node ordering on the x-axis.

address this burden, national institutions such as NIH have included brain-behavior research as one of their strategic objectives and stated that sound efforts must be made to redefine mental disorders into dimensions or components of observable behaviors that are more closely aligned with the biology of the brain. Using brain imaging data to predict diagnosis has great potential to result in mechanisms that target for more effective preemption and treatment.

In this paper, we present BRAIN NETWORK TRANSFORMER, a specialized graph Transformer model with ORTHONORMAL CLUSTERING READOUT for brain network analysis. Extensive experiments on two large-scale brain network datasets demonstrate that our BRAINNETTF achieves superior performance over SOTA baselines of various types. Specifically, to model the potential node feature similarity in brain networks, we design OCREAD and prove its effectiveness both theoretically and empirically. Lastly, the re-standardized dataset split for ABIDE can provide a fair evaluation for new methods in the community. For future work, BRAINNETTF can be improved with explicit explanation modules and used as the backbone for further brain network analysis, such as digging essential neural circuits for mental disorders and understanding cognitive development in adolescents.

## 6 Acknowledgments

This research was supported in part by the University Research Committee of Emory University, and the internal funding and GPU servers provided by the Computer Science Department of Emory University. The authors gratefully acknowledge support from NIH under award number R01MH105561 and R01MH118771. The content is solely the responsibility of the authors and does not necessarily represent the official views of the National Institutes of Health.

Data used in the preparation of this article were obtained from the Adolescent Brain Cognitive Development (ABCD) Study (`https://abcdstudy.org`), held in the NIMH Data Archive (NDA). This is a multisite, longitudinal study designed to recruit more than 10,000 children age 9-10 and follow them over 10 years into early adulthood. The ABCD Study® is supported by the National Institutes of Health and additional federal partners under award numbers U01DA041048, U01DA050989, U01DA051016, U01DA041022, U01DA051018, U01DA051037, U01DA050987, U01DA041174, U01DA041106, U01DA041117, U01DA041028, U01DA041134, U01DA050988, U01DA051039, U01DA041156, U01DA041025, U01DA041120, U01DA051038, U01DA041148, U01DA041093, U01DA041089, U24DA041123, U24DA041147. A full list of supporters is available at `https://abcdstudy.org/federal-partners.html`. A listing of participating sites and a complete listing of the study investigators can be found at `https://abcdstudy.org/consortium_members/`. ABCD consortium investigators designed and implemented the study and/or provided data but did not necessarily participate in the analysis or writing of this report. This manuscript reflects the views of the authors and may not reflect the opinions or views of the NIH or ABCD consortium investigators. The ABCD data repository grows and changes over time. The ABCD data used in this report came from NIMH Data Archive Release 4.0 (DOI 10.15154/1523041). DOIs can be found at `https://nda.nih.gov/abcd`.

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
