# A  Training Curves of Different Models with or without StratifiedSampling

In Figure 5, we demonstrate the training curves of different models with or without stratified sampling based on site information from ABIDE. The curves of different variants display similar patterns across three model architectures in a single run. We remove Graphormer since its performance is much worse than others. Specifically, it is shown that (a) with stratified sampling, the performance gap between validation and test on ABIDE is much smaller than the one without stratified sampling; (b) stratified sampling can stabilize the training process on ABIDE, especially for VanillaTF and BRAINNETTF.

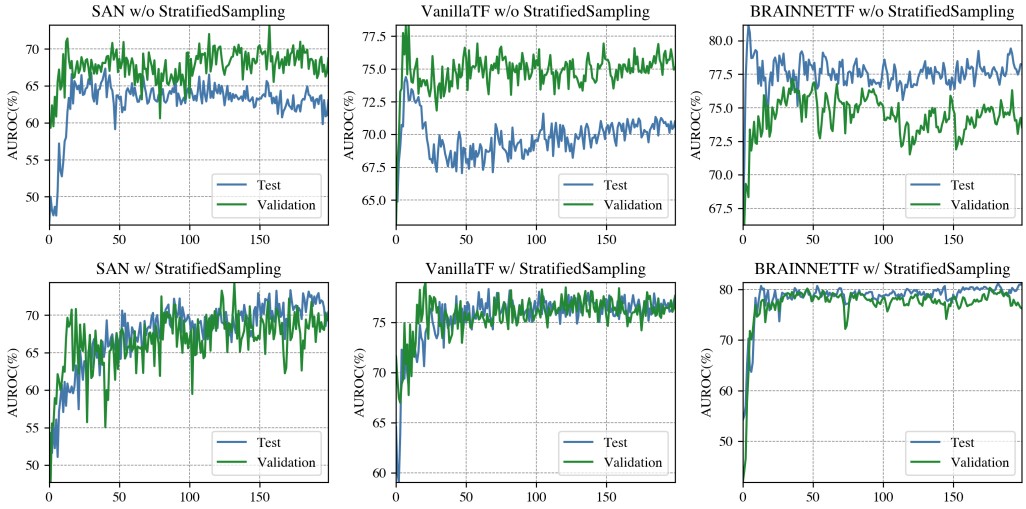

Figure 5: Training Curves of Different Models with or without StratifiedSampling.

# B  Transformer Performance with Different Node Features

We compare the performance of Transformer model equipped with different node features. The results are shown in Table 3, where connection profile represents the corresponding row for each node in the adjacency matrix, identity feature initializes a unique one-hot vector for each node, and eigen feature generates a $k$-dimensional feature vector for each node from the $k$ eigenvectors based on the eigendecomposition on the adjacency matrix. Empirical observations demonstrate that adding identity or eigen node features to connection profiles cannot improve the model's performance.

| Model | Node Feature | Dataset | |
| --- | --- | --- | --- |
| | | ABIDE | ABCD |
| | Connection Profile | 76.4±1.2 | 94.3±0.7 |
| VanillaTF | Connection Profile w/ Identity Feature | 75.4±1.9 | 94.5±0.6 |
| | Connection Profile w/ Eigen Feature | 75.9±2.1 | 94.0±0.8 |

Table 3: The Performance (AUROC%) of Transformer with Different Node Features.

# C  Statistical Proof of the Goodness with Orthonormal Cluster Centers

We propose two statistical methods to prove the goodness in orthonormal case since it is impractical to directly compare the performance of the orthonormal and non-orthonormal initializations.

## C.1 Proof of Theorem 3.1

We state Theorem 3.1 here and show the proof details.

**Theorem C.1.** *For arbitrary $r > 0$, let $B_r = \{\mathcal{Z} \in \mathbb{R}^V; \|\mathcal{Z}\| \leq r\}$ denote the round ball centered at origin of radius $r$ with $\mathcal{Z}$ being fracture vectors. Let $V_r$ be the volume of $B_r$. The variance of Softmax projection averaged over $B_r$*

$$\frac{1}{V_r} \int_{B_r} \sum_k^K \left( \frac{e^{\langle \mathcal{Z}, \boldsymbol{E}_{k\cdot} \rangle}}{\sum_{k'}^K e^{\langle \mathcal{Z}, \boldsymbol{E}_{k'\cdot} \rangle}} - \frac{1}{K} \right)^2 d\mathcal{Z}, \tag{6}$$

*attains maximum when $\boldsymbol{E}$ is orthonormal.*

*Proof.* For simplicity, we first consider the two-dimensional case with two cluster centers $\boldsymbol{E}_1, \boldsymbol{E}_2$. Since we integrate over the round ball $B_r$, spherical symmetry allows us to set $\boldsymbol{E}_1 = (1, 0)$ and $\boldsymbol{E}_1 = (\cos(\phi), \sin(\phi))$ with $\phi \in [0, \frac{\pi}{21}]$ being the angle between $\boldsymbol{E}_1$ and $\boldsymbol{E}_2$ under polar coordinates. Then the Softmax readout Eq. (2) can be rewritten as:

$$\boldsymbol{P}_1 = \frac{e^{\rho \cos(\theta)}}{e^{\rho \cos(\theta)} + e^{\rho \cos(\theta - \phi)}}, \quad \boldsymbol{P}_2 = \frac{e^{\rho \cos(\theta - \phi)}}{e^{\rho \cos(\theta)} + e^{\rho \cos(\theta - \phi)}}, \tag{7}$$

where $\theta$ is the angle between $\mathcal{Z}$ and $\boldsymbol{E}_1$ and $\rho$ is the norm of $\mathcal{Z}$. Hence, the integral is

$$F(\phi) := \frac{1}{V_r} \int_{B_r} \sum_{k=1}^2 (\boldsymbol{P}_k - \frac{1}{2})^2 d\mathcal{Z} = \frac{1}{\pi r^2} \int_0^r \int_0^{2\pi} \left( \frac{e^{2\rho \cos(\theta)} + e^{2\rho \cos(\theta - \phi)}}{(e^{\rho \cos(\theta)} + e^{\rho \cos(\theta - \phi)})^2} + \frac{1}{2} \right) d\theta d\rho. \tag{8}$$

Our aim is to show that the integral $F(\phi)$ attains its maximum when $\boldsymbol{E}_1, \boldsymbol{E}_2$ are orthogonal. It is unclear whether the above integral has an elementary antiderivative. Thus, instead of evaluating the integral directly, we firstly prove two symmetric properties of the integrand $f(\rho, \theta, \phi)$: (a) It is straightforward to show that $f(\rho, \theta + k\pi, \phi) = f(\rho, \theta, \phi)$ for $k \in \mathbb{N}$. That is, $f$ is periodic for $\pi$ on the first argument $\theta$. (b) We have

$$
\begin{aligned}
f(\frac{\phi}{2} + \frac{\pi}{2} - \theta) &= \frac{e^{2\rho \sin(\frac{\phi}{2} + \theta)} + e^{-2\rho \sin(\frac{\phi}{2} - \theta)}}{(e^{\rho \sin(\frac{\phi}{2} + \theta)} + e^{-\rho \sin(\frac{\phi}{2} - \theta)})^2} \\
&= \frac{e^{2\rho \sin(\frac{\phi}{2} + \theta)} + e^{-2\rho \sin(\frac{\phi}{2} - \theta)}}{e^{2\rho \sin(\frac{\phi}{2} + \theta)} + e^{-2\rho \sin(\frac{\phi}{2} - \theta)} + 2e^{\rho \sin(\frac{\phi}{2} + \theta) - \rho \sin(\frac{\phi}{2} - \theta)}} \\
&= \frac{e^{2\rho \sin(\frac{\phi}{2} - \theta)} + e^{-2\rho \sin(\frac{\phi}{2} + \theta)}}{(e^{\rho \sin(\frac{\phi}{2} - \theta)} + e^{-\rho \sin(\frac{\phi}{2} + \theta)})^2} = f(\frac{\phi}{2} + \frac{\pi}{2} + \theta),
\end{aligned}
\tag{9}
$$

which means $f$ is symmetric with respect to $\theta = \frac{\phi}{2} + \frac{\pi}{2} + k\pi$. As the integrand $f(\rho, \theta, \phi)$ is periodic, we are allowed to compare $F(\phi_1), F(\phi_2)$ via

$$
\begin{aligned}
\int_{\frac{\phi_1}{2}}^{\frac{\phi_1}{2} + 2\pi} f(\rho, \theta, \phi_1) d\theta &= \int_0^{2\pi} f(\rho, \theta, \phi_1) d\theta, \\
\int_{\frac{\phi_1}{2}}^{\frac{\phi_2}{2} + 2\pi} f(\rho, \theta, \phi_2) d\theta &= \int_0^{2\pi} f(\rho, \theta, \phi_2) d\theta.
\end{aligned}
\tag{10}
$$

The integral domain $[\frac{\phi}{2}, \frac{\phi}{2} + 2\pi]$ is taken according to the second symmetry property of $f$ and can be significant for the following trick: we take the directional derivative of $f$ along $\boldsymbol{v} = (1, 2)$ tangent to the straight line $\theta = \frac{\phi}{2}$:

$$
\begin{aligned}
Df(\boldsymbol{v}) &= \frac{\partial f}{\partial \theta} + 2 \frac{\partial f}{\partial \phi} \\
&= \frac{2\rho e^{\rho \cos(\theta - \phi) + \rho \cos(\theta)} (e^{\rho \cos(\theta - \phi)} - e^{\rho \cos(\theta)})(\sin(\theta) + \sin(\theta - \phi))}{(e^{\rho \cos(\theta - \phi)} + e^{\rho \cos(\phi)})^3}.
\end{aligned}
\tag{11}
$$

It is easy to check that in the above integral domain and for any $\rho > 0$, $Df(\boldsymbol{v})$ is always non-negative. Hence,

$$f(\rho, \theta - \frac{\phi_1}{2}, \phi_1) \leq f(\rho, \theta - \frac{\phi_2}{2}, \phi_2) \tag{12}$$

when $\phi_1 \leq \phi_2$. After taking integral, $F(\phi_1) \leq F(\phi_2)$ and thus it attains maximum in the orthonormal case ($\phi = \frac{\pi}{2}$). Comparing $F(\phi_1), F(\phi_2)$ without adjusting the integral domain as above cannot give a clear result because the simple partial derivative $\partial f / \partial \phi$ oscillates around zero. Higher dimensional cases follow similarly by employing spherical and hyperspherical coordinates. $\qquad\square$

### C.2 Proof of Theorem 3.2

Theorem 3.2 deals with a more general case: comparing the performance of an arbitrary readout $\boldsymbol{P}$ defined by orthonormal cluster centers with non-orthonormal ones. We regard $\boldsymbol{P}$ as an estimated similarity probability between nodes and clusters and solve this problem from the perspective of statistics. The estimation is considered as a regression of samples $(\hat{\boldsymbol{Z}}^{(s)}, \hat{\boldsymbol{E}}^{(t)}, \hat{\boldsymbol{P}}^{(st)})$ from node features, cluster centers and similarity probabilities. We then judge the estimation relative to true similarity probability $\boldsymbol{P}_T$. Although it is almost impossible to find an analytic formula for $\boldsymbol{P}_T$, we can indirectly judge the quality of estimation. To clarify the idea, we introduce some basic concepts from statistics and prove our results on a statistical basis.

#### C.2.1 Background Knowledge of Regression Analysis

We first consider process samples by logistic regression with cluster centers as *categorical variables*. Intuitively, non-orthonormal centers correlate with each other, which means there is an *overlap* among categorical variables and makes it hard to identify the *decision boundary* that leads to a failed classification. However, as far as we know, it is *unclear* how to compare overlaps between orthonormal and non-orthonormal variables rigorously. Thus, we simply process samples by a general nonlinear regression. The regression process is linearized by the Gauss-Newton algorithm to facilitate the analysis. We judge the *goodness-of-fit* describing the degree to which the regression function fits its observed value, and then conduct a hypothesis test. The *goodness-of-fit* is measured by *coefficient of determinate $R^2$* [47]:

**Definition C.2.** We consider a regression with $r$ independent main variables:

$$Y = \beta_0 + \beta_1 X_1 + \beta_2 X_2 + \cdots + \beta_r X_r + \epsilon. \tag{13}$$

Let $\hat{x}_p = (\hat{x}_{p1}, ..., \hat{x}_{ps})^\top$ and $\hat{y} = (\hat{y}_1, ..., \hat{y}_s)^\top$ be data sets (samples) associated with *fitted values* $\check{y} = (\check{y}_1, ..., \check{y}_s)$. Each difference $e_q = \hat{y}_q - \check{y}_q$ is called a *residue*. We denote the mean of $\hat{x}_p$ and $\hat{y}$ by $\bar{x}_p, \bar{y}$. The variability of data set can be measured by the *total sum of squares* (SST), the *sum of squares of residuals* (SSR) and the *explained sum of squares* (SSE) defined as (where $p = 1, 2, ..., r$ $q = 1, 2, ..., s$):

$$\text{SST} = \sum_q (\hat{y}_q - \bar{y})^2, \quad \text{SSR} = \sum_q e_q^2 = \sum_q (\hat{y}_q - \check{y}_q)^2, \quad \text{SSE} = \sum_{q,p} (\hat{x}_{qp} - \bar{x}_p)^2. \tag{14}$$

In linear regression, $\text{SSR} + \text{SSE} = \text{SST}$ and the coefficient of determination $R^2$ is defined as:

$$R^2 = \frac{\text{SSE}}{\text{SST}} = 1 - \frac{\text{SSR}}{\text{SST}}. \tag{15}$$

Conceptually, SSE is the error cost by regression of main variables. Thus by definition, $R^2$ reveals the percentage of errors that main variables can explain in the total error SST. The value of $R^2$ is bounded by 1. A large value of $R^2$ indicates a better fitting. However, it should be noted that an extremely-large $R^2$ could indicate overfitting.

In our problem, since our regression is nonlinear, the sum of SSR and SSE is less than SST [1]. Therefore, measuring *goodness-of-fit* by $R^2$ in nonlinear regression is inaccurate. A common strategy to remedy this problem is approximating nonlinear functions by polynomials via *Gauss-Newton algorithm*. We provide a brief introduction here, and more details can be found in [1]: for a nonlinear

model $f_k$ with parameter $\delta$, in a small neighborhood of $\delta_T$-the true value of $\delta$, we have the linear expansion:

$$f_k(\delta) \approx f_k(\delta_T) + \sum_{m=1}^{M} \frac{\partial f_k}{\partial \delta_m}\bigg|_{\delta_T} (\delta_m - \delta_{Tm}). \tag{16}$$

Or briefly, we write it by *vector notation*:

$$\boldsymbol{f}(\delta) \approx \boldsymbol{f}(\delta_T) + \boldsymbol{F}(\delta - \delta_T), \tag{17}$$

where $\boldsymbol{F}(\delta - \delta_T)$ stands for the dot product of derivatives and differences of parameters from Eq. (16). Suppose $\delta^{(\gamma)}$ is an approximation to the least-squares estimation $\delta$ of our model, for $\delta$ close to $\delta^{(\gamma)}$, we rewrite the expansion as:

$$\check{\boldsymbol{P}} = \boldsymbol{f}(\delta) \approx \boldsymbol{f}(\delta^{(\gamma)}) + \boldsymbol{F}^{(\gamma)}(\delta - \delta^{(\gamma)}), \tag{18}$$

where $\check{\boldsymbol{P}}$ denotes a fitted value of $\boldsymbol{P}$ and $F^{(\gamma)}(\delta - \delta^{(\gamma)})$ again means a dot product. Applying this to the residual vector $\boldsymbol{e}(\delta)$, we have:

$$\boldsymbol{e}(\delta) = \boldsymbol{P} - \boldsymbol{f}(\delta) \approx \boldsymbol{e}(\delta^{(\gamma)}) - \boldsymbol{F}^{(\gamma)}(\delta - \delta^{(\gamma)}). \tag{19}$$

Thus, the norm

$$\begin{aligned}
S(\delta) &:= \|\boldsymbol{P} - f(\delta)\|^2 = \boldsymbol{e}^\top(\delta)\boldsymbol{e}(\delta) \\
&\approx \boldsymbol{e}^\top(\delta^{(\gamma)})\boldsymbol{e}(\delta^{(\gamma)}) - 2\boldsymbol{e}^\top(\delta^{(\gamma)})\boldsymbol{F}^{(\gamma)}(\delta - \delta^{(\gamma)}) + (\delta - \delta^{(\gamma)})^\top \boldsymbol{F}^{(\gamma)\top}\boldsymbol{F}^{(\gamma)}(\delta - \delta^{(\gamma)}).
\end{aligned} \tag{20}$$

The right-hand side is minimized with respect to $\delta$ when

$$\delta - \delta^{(\gamma)} = (\boldsymbol{F}^{(\gamma)\top}\boldsymbol{F}^{(\gamma)})^{-1}\boldsymbol{F}^{(\gamma)\top}\boldsymbol{e}(\delta^{(\gamma)}) = \zeta^{(\gamma)}. \tag{21}$$

This suggests that given a current approximation $\delta^{(\gamma)}$, the next approximation should be:

$$\delta^{(\gamma+1)} = \delta^{(\gamma)} + \zeta^{(\gamma)}. \tag{22}$$

Expanding the nonlinear function $\boldsymbol{f}$ as polynomials and modifying the parameter $\delta$ as above, we can use $R^2$ to measure the *goodness-of-fit*. To acquire higher accuracy in a general nonlinear regression, one can make a elaborated *goodness-of-fit test* for specific fitting functions e.g., [9, 11]. We do not discuss this sophisticated method as it is out of the scope of this paper.

### C.2.2 Comparing $R^2$ by Variance Inflation Factor

The proof of Theorem 3.2 consists of two steps: (a) we first prove that the *regression accuracy*, the accuracy when regressing $\boldsymbol{P}$ is higher when sampling from orthonormal cluster centers (Theorem C.4), and consequently (b) higher regression accuracy increases *appraisal accuracy*, the accuracy when appraising an estimated value in *hypothesis testing* (Theorem C.6).

In this subsection, we compare regression accuracy. we fix $\boldsymbol{Z}_i$ when regressing $\boldsymbol{P}$ via the fitted value $\check{\boldsymbol{P}}(\boldsymbol{E}_k)$. Statistically, the expectation $E(\boldsymbol{P})$ of all readouts is identified as the true similarly probability $\boldsymbol{P}_T$. In regression analysis, the Ordinary Least Squares (OLS) guarantees asymptotically unbiased estimations. That is, when the sample size $s$ is large enough, it can be regarded as an *unbiased estimation* [47]:

$$E(\check{\boldsymbol{P}}) = \boldsymbol{P}_T = E(\boldsymbol{P}). \tag{23}$$

Therefore, the better the *goodness-of-fit* reflected by $R^2$, the smaller the variance of estimation. To compare this, we use the concept of *variance inflation factor* which reflects the inflation of weights of variables in regression:

**Definition C.3.** The variance inflation factor $(\text{VIF})_p$ is defined as:

$$(\text{VIF})_p = \frac{1}{(1 - R_p^2)}, \tag{24}$$

where $R_p^2$ is the coefficient of multiple determination when $X_p$ is regressed by the r-1 other variables in the model from Eq. (13).

*Remark.* We discuss more details about VIF in the following context [47]. For simplicity, we denote the following collection of samples and regression coefficients:

$$\hat{X} = (\hat{x}_1, ..., \hat{x}_r) = (\hat{x}_{qp}), \quad \hat{y} = (\hat{y}_1, ..., \hat{y}_s)^\top, \quad \beta = (\beta_1, ..., \beta_r).$$

In the regression model Eq. (13), the estimation $\check{\beta}_p$ of regression coefficients $\beta_p$ are obtained by Ordinary Least Squares (OLS):

$$\check{\beta} = (\hat{X}^\top \hat{X})^{-1} \hat{X}^\top \hat{y}. \tag{25}$$

We standardize the regression equation by covariance matrices $\sigma_y$ of $\check{y}$ and the variance $\sigma_q$ of $\hat{x}_p$ as

$$\check{y}_q^* = \frac{\check{y}_q - \bar{y}}{\sigma_y}, \quad \hat{x}_{qp}^* = \sigma_q^{-1}(\hat{x}_{pq} - \bar{x}_p), \tag{26}$$

and

$$\check{\beta}_q^* = \check{\beta}_q \frac{\sigma_q}{\sigma_y}, \quad \check{y}^* = \check{\beta}_0^* + \check{\beta}_1^* X_1^* + \check{\beta}_2^* X_2^* + \cdots + \check{\beta}_r^* X_r^*. \tag{27}$$

Similarly to Eq. (25), standardized estimation of regression coefficients are equal to

$$\check{\beta}^* = (\check{X}^{*\top} \check{X}^*)^{-1} \check{X}^{*\top} \check{y}^*. \tag{28}$$

On the other hand, the covariance matrix of the estimated regression coefficients is

$$\sigma_{\check{\beta}}^2 = \sigma^2 (X^\top X)^{-1}, \quad \sigma^2 = \sum_{q=1}^{s} (\check{y}_q - \bar{y})^2, \tag{29}$$

where $\sigma^2$ is the *error term variance* for $X$ (cf. Definition C.2). After standardization, it is noted that $X^{*\top} X^*$ is just the correlation matrix $r_{XX}$ of $X^*$. Hence, by Eq. (29) we obtain:

$$\sigma_{\check{\beta}^*}^2 = (\sigma^*)^2 r_{XX}^{-1}. \tag{30}$$

Let $(\text{VIF})_p$ be the $p$-th diagonal element of the matrix $r_{XX}^{-1}$. The variance of $\beta_p^*$ is equal to:

$$\sigma_{\check{\beta}_p^*}^2 = (\sigma^*)^2 (\text{VIF})_p. \tag{31}$$

The diagonal element $(\text{VIF})_p$ is just the variance inflation factor for $\check{\beta}_p^*$. The variance of $\beta_p^*$ can also be written as [47]

$$\sigma_{\check{\beta}_p^*}^2 = \frac{1}{1 - R_p^2} \left[ \frac{\sigma^2}{\sum_q (x_{qp} - \bar{x}_p)^2} \right]. \tag{32}$$

With the previous discussion, we conclude that

$$(\text{VIF})_p = \frac{1}{(1 - R_p^2)}, \tag{33}$$

where $R_p^2$ is defined in C.3.

**Theorem C.4.** *Let*

$$VIF = \frac{\sum_{p=1}^{r} (VIF)_p}{r - 1}, \tag{34}$$

*where $r$ denotes the number of variables in Eq. (13). Then* VIF $\geq$ *1 with equality holds if and only if the variables are orthogonal.*

*Proof.* To prove this, we need to generalize the definition of $R^2$. By definition,

$$R^2 = \frac{\text{SSE}}{\text{SST}} = \frac{\sum_{q=1}^{s} (\check{y}_q - \bar{y})^2}{\sum_{q=1}^{s} (y_q - \bar{y})^2} = \sum_{q=1}^{s} (\check{y}_q^*)^2. \tag{35}$$

Substituting Eq. (27) into the above identity, we have

$$\sum_{q=1}^{s}(\check{y}_q^*)^2 = \sum_{q=1}^{s}(\check{X}_q^*\check{\beta}^*)^2 = (X_q^*\check{\beta}^*)^\top X_q^*\check{\beta}^*, \tag{36}$$

and by Eq. (28), we conclude that

$$R^2 = (r_{XY})^\top (r_{XX})^{-1} r_{XY}. \tag{37}$$

As the finial step, we compute $R_p^2$ from Definition C.3 by Eq. (37). It should be noted that according to Definition C.3, $R_p^2$ is the *goodness-of-fit* when $X_p$ is regressed by the r-1 other variables. These variables are uncorrelated in orthonormal case. Hence $r_{XY} = 0, R_p^2 = 0$ and VIF $= 1$. □

*Remark.* In statistics, when a variable's VIF is greater than 1, or equivalently $R_p^2 \neq 0$, the influence of this variable on the whole estimation is inflated. It breaks the so-called *absence of multicollinearity*, a fundamental principle in multiple regression analysis, and hence causes more error. Since SSE is a constant value, the error generated by the inflation would be counted into SSR, which leads to a decrease in $R^2$ by Definition C.2 (see [47, 1] for more details).

### C.2.3    Statistical Hypothesis Testing

The previous discussion verifies that regressing with orthonormal samples attains a higher *goodness-of-fit*. In other words, it achieves a higher regression accuracy. Tools from *hypothesis testing* are borrowed here to determine the appraisal accuracy mentioned at the beginning of Section C.2.2. We first introduce *mean squared error* (MSE) commonly used in statistics [19]:

**Definition C.5.** Recall that the residue $e_q = (\hat{y}_q - \check{y}_q)$ from Definition C.2. Then,

$$\text{MSE} = \frac{1}{s}\sum_{q=1}^{s}(\hat{y}_q - \check{y}_q)^2 = \frac{1}{s}\sum_{q=1}^{s}(e_q)^2 = \frac{1}{s}e^\top e. \tag{38}$$

As mentioned in C.2.1, a small coefficient of determination $R^2$ indicates a large SSR and hence leads to a large MSE. As a result of Theorem C.4, MSE is minimized in the orthonormal case.

We now assume a domain centered at the true value $P_T$ of radius $d$, and treat the outside space $W$ as the *rejection region*. Statistically, if the distance between $\check{P}$ and $P_T$ is less than a small enough $d$, we can regard them as the same. Intuitively, if fitted values $\check{P}$ are largely scattered from the true value $P_T$, that is, when MSE is large, it can interfere with our judgment of whether $P$ can be identified with $P_T$. Rigorously, we make a *hypothesis testing* and analyze the probability of rejecting a well-estimated readout function. We prove in the following that when sampling from orthonormal cluster centers, a higher regression accuracy (Theorem C.4) guarantees a lower MSE and therefore increases the appraisal accuracy.

**Theorem C.6.** *The significance level $\alpha_{E_k}$ reveals that the probability of rejecting a well-estimated readout is lower when sampling from orthonormal centers than sampling from non-orthonormal centers.*

*Proof.* Let $P$ be a readout function such that $\|P_T - P\| \leq d$ for small enough $d$. Statistically, we can treat them as the same and simply write $\check{P} = P_T$. In *hypothesis testing*, we define *null hypothesis* $H_0$ and *alternative hypothesis* $H_1$ by

$$H_0 : \check{P} = P_T, \quad H_1 : \check{P} \neq P_T, \tag{39}$$

in which $H_1$ means that we reject a well-estimated readout with $H_0$ having the opposite meaning. The rejection region for this test is thus given as $W = \{\check{P} \neq P_T\}$. As a conventional procedure in *hypothesis testing*, we take a suitable test statistic $T_{E_k}(Z_i)$ whose distribution $f$ is known [19]. It is used to compute the probability that $\check{P}$ is in the rejection region. The corresponding probability distribution is called potential function $g(\theta)$ for $W$ in this setting:

$$g(\theta) = P_\theta(\check{P} \in W) = \int_W f(T_{E_k}(Z_i))dZ_i \leq \alpha_{E_k}, \quad \theta = H_0 \cup H_1, \tag{40}$$

where the significance level $\alpha_{\boldsymbol{E}_k}$ is the upper bound of the probability of making mistakes (formally called *type I error*) [19].

By Theorem C.4 and Remark C.5, MSE is minimized in the orthonormal case. It can be treated as a variance of distribution $f$. Then by *Vysochanskij–Petunin inequality*, a refinement of Chebyshev inequality, the integration over $W$ with orthonormal cluster centers $\boldsymbol{E}_k$ is smaller than that with non-orthonormal cluster centers $\boldsymbol{E}'_k$:

$$\int_W f(T_{\boldsymbol{E}_k}(\boldsymbol{Z}_i))d\boldsymbol{Z}_i \leq \int_W f(T_{\boldsymbol{E}'_k}(\boldsymbol{Z}_i))d\boldsymbol{Z}_i. \tag{41}$$

As the result holds true for any well-chosen $T_{\boldsymbol{E}_k}(\boldsymbol{Z}_i)$, $\alpha_{\boldsymbol{E}_k} \leq \alpha_{\boldsymbol{E}'_K}$, this finishes the proof. $\qquad\square$

# D    Running Time

Table 4 shows that state-of-the-art models of Graphormer and SAN are much slower than our BRAINNETTF and VanillaTF, mainly because their implementations are not optimized toward the unique properties of brain networks. Specifically, let $e$ be the number of edges and $v$ be the number of nodes. The calculation of Graphormer and SAN optimizes the case where $e \ll v^2$. However, brain networks usually have a small number of nodes but dense connections, i.e., $e \simeq v^2$. Therefore the optimized sparse graph operations in PyTorch Geometric [23] do not work properly. On the other hand, since the number of nodes in brain networks is usually relatively small (less than 500), we can directly speed up the calculation using matrix multiplication, which is what we did in BRAINNETTF and VanillaTF. Besides, the edge feature generation operator in Graphormer further increases the burden on its computing time.

Table 4: Running time with different graph transformer methods.

| Method | Running Time on ABIDE (min) | Running Time on ABCD (min) |
|---|---|---|
| SAN | 93.01±0.96 | 908.05±3.6 |
| Graphormer | 133.52±0.54 | 4089.86±5.7 |
| VanillaTF | 2.32±0.10 | 36.26±2.12 |
| BRAINNETTF | **1.98±0.04** | **30.31±1.16** |

# E    Number of Parameters

Table 5: The number of parameters in different models.

| Method | #Para on ABIDE | #Para on ABCD |
|---|---|---|
| BrainNetCNN | 0.93M | 0.93M |
| BrainGB | 1.08M | 1.49M |
| FBNetGen | 0.55M | 1.18M |
| SAN | 57.7M6 | 186.7M |
| Graphormer | 1.23M | 1.66M |
| VanillaTF | 15.6M | 32.7M |
| BRAINNETTF | 4.0M | 11.2M |

# F    Parameter Tuning

For BrainGB, BrainGNN, FBNetGen, we use the authors' open-source codes. For SAN and Graphormer, we folk their repositories and modified them for the brain network dataset. For BrainNetCNN and VanillaTF, we implement them by ourselves. We use the grid search for some important hyper-parameters for these baselines based on the provided best setting. To be specific, for BrainGB, we search different readout functions {mean, max, concat} with different message-passing functions {Edge weighted, Node edge concat, Node concat}. For BrainGNN, we search different learning rates {0.01, 0.005, 0.001} with different feature dimensions {100, 200}. For FBNetGen, we search different encoders {1D-CNN, GRU} with different hidden dimensions {8, 12, 16}. For

BrainNetCNN, we search different dropout rates {0.3, 0.5, 0.7}. For VanillaTF, we search the number of transformer layers {1, 2, 3} with the number of headers {2, 4, 6}. For SAN, we test LPE hidden dimensions {4, 8, 16}, the number of LPE and GT transformer layers {1, 2} and the number of headers {2, 4} with 50 epochs training. For Graphormer, we test encoder layers {1, 2} and embed dimensions {256, 512}. Furthermore, since the rebuttal time is pretty short, we do not have enough time to dig two new baselines, BrainnetGNN and DGM, which may be why their performance is worse than others.

## G   Software Version

Table 6: The dependency of BRAINNETTF.

| Dependency | Version |
|:---:|:---:|
| python | 3.9 |
| cudatoolkit | 11.3 |
| torchvision | 0.13.1 |
| pytorch | 1.12.1 |
| torchaudio | 0.12.1 |
| wandb | 0.13.1 |
| scikit-learn | 1.1.1 |
| pandas | 1.4.3 |
| hydra-core | 1.2.0 |

## H   The Difference between Various Initialization Methods

To show orthonormal initialization can produce more discriminative $P$ between classes than random initialization, we calculate the difference score $d$ based on the formula

$$d = \sum_{i}^{K} \sum_{j}^{V} \frac{|P_{ij}^{female} - P_{ij}^{male}|}{KV}, \tag{42}$$

where $V$ is the number of nodes and $K$ is the number of clusters. After running the t-test, we found the margins between random and orthonormal on both ABIDE and ABCD are significant, which is consistent with our conclusion.

Table 7: The difference score between different initialization methods.

| Method | Difference score on ABIDE | Difference score on ABCD |
|:---:|:---:|:---:|
| Random | 0.067±0.016 | 0.125±0.010 |
| Orthonormal | 0.085±0.015 | 0.142±0.014 |