# OpenReview forum: "Brain Network Transformer"
_NeurIPS.cc/2022/Conference — NeurIPS 2022 Accept_

### Official Review · Reviewer_xfsP · 2022-07-10

**Rating:** 7
**Confidence:** 4
**Soundness:** 3 good
**Presentation:** 3 good
**Contribution:** 3 good

**Summary:**

The paper introduces a new transformer-based model for brain network analysis bringing better improvements on the ABIDE and ABCD datasets when compared to other transformer-based models. The model's architecture is motivated on insights from two previous papers [12, 25] to decide on the initial node features and to learn graph structures to achieve superior effectiveness. The paper makes a further contribution of providing a re-standardisation of an evaluation pipeline for the ABIDE dataset, given findings on unstable training.


--------------------------------------
[UPDATED AFTER REBUTTAL DISCUSSIONS]
In my opinion, the authors tackled my question in a satisfactory way, and I am making these changes to my review:

Soundness: 2 > 3
Contribution: 2 > 3
Rating: 5 > 7

**Questions:**

The five points in "Major weaknesses" are the ones that made me not give a higher score, therefore I believe those are the points which could change my opinion the most on the scores given.

**Limitations:**

The paper contains predictions using human brain data, with some labels being sex and diagnosis. Therefore, I find it incorrect that the paper does not follow the recommendations on writing about societal impact when the prediction of these labels in specific can have so many different impacts in society.  I believe the acknowledgement of these variables would be the bare minimum needed for a NeurIPS paper (maybe starting from the last sentence of conclusion).

With regards to limitations, I believe they were tackled properly in Conclusion and in other sections of the paper. However, as a consequence of the answers to my questions this will be be better after the rebuttal process.

**Strengths And Weaknesses:**

Overall I think this is a good, well-written paper. Despite being yet another paper exploring transformer-based models, the results seem to be sound and the ablation analysis goes one step further and shows that the OCRead module can be useful in some other architectures. The proofs in supplementary material are a good addition and I thank the inclusion of a link for me to analyse the code during this review process. All these strengths are enough for me to recommend (borderline) acceptance, but I have some important concerns that make me not give a higher score at the moment, which I hope the authors can answer during the rebuttal period. I will first highlight what are my major concerns/doubts, and then leave a list of some other less important comments/suggestions.

Before going into it, **I must ask the authors to please double-check their terms to anonymise in the code shared** because while I was analysing the code, I found at least one `.yaml` file under the `setting` folder with an absolute path. It was quite obvious to me that it contained a username resembling quite closely some important papers referenced so I stopped looking more into it, but still would ask the authors to correct this in case we need to open the anonymised repo during the rebuttal process.

## Major weaknesses

1. The paper cites Kan et al. [25] to justify the idea of having the network to learn the graph structure, and uses the model presented in this reference (FBNETGEN) as another baseline for comparison. Although I know that choosing baselines can be a very subjective topic, I must say that selecting only one baseline for a learnable brain network is not enough. I have seen at least two possible papers learning a graph structure (arXiv:2002.04999, 10.1016/j.media.2020.101709) which were never evaluated on ABIDE/ABCD and could therefore be another interesting contribution of this paper to use this previous literature on unexplored datasets, even though the 2 papers I leave here had some analysis on medical images and were not used in the original FBNETGEN paper. From Table 1, I noticed that BrainGB and FBNETGNN seem to come from the same group as they come from the two references used to motivate the some choices on the model (refs 12 and 25), which is another non-subjective reason to include another baseline from another paper. Finally, the "learnable brain network" type in Table 1 is the one with the closest performance to BrainNetTF and therefore another baseline of the same type would improve the significance of this model. For all these reasons, I believe the authors need to add at least another meaningful baseline on the "learnable brain network" category in order for their results to be more sound.
2. I believe the authors are mistakenly using "gender" when they mean biological sex. As far as I know, these datasets do not provide information on the gender of the people in the datasets, but rather the biological binary sex. I therefore request the authors to correct this for example in figure 2 and in the other occurrences of "gender".
3. On lines 170-171 the authors claim that "none of the existing readout functions leverages the properties of brain networks that nodes in the same functional modules tend to have similar behaviors and clustered representations". After reading this sentence I immediately thought about the DiffPool paper (arXiv:1806.08804) and so many other papers adapted from this differentiable graph pooling module which, in practice, forms different clusters as a way to pool the nodes representations. I know of at least one recent paper (10.1016/j.media.2022.102471) using it in the medical context as readout function, so I believe the authors should probably downtown this sentence, and find a more complex (and who knows fairer) readout/pooling operator to compare OCRead with. From what I see in the code, the authors used DGL with pytorch, which I believe has implementations available for more complex pooling layers (https://docs.dgl.ai/en/0.8.x/api/python/nn-pytorch.html#global-pooling-layers) or DiffPool (https://github.com/dmlc/dgl/blob/master/examples/pytorch/diffpool/model/tensorized_layers/diffpool.py).
4. I do not agree with the paper's justification in lines 270-272 that AUROC is the most proper performance metric, given for example that in the medical context a model needs to use a single threshold to decide on the positive/negative output, and AUROC tests all thresholds in practice. Besides, the model is trained such that the highest AUROC performance on the validation set is attained, which could artificially inflate these results and we do not know how they affect other very important metrics in the medical context like sensitivity, specificity, or even model calibration. I request the authors to at least add values on sensitivity and specificity (or equivalent) to show the strengths of this work.
5. In Table 1 I find it difficult to understand how fair the comparisons are. Are these models similar in terms of number of layers? Or number of parameters? Given BrainNetTF seems to have had some sort of hyperparameter search, it is probably unfair to compare with out-of-the-box models?


## Other small weaknesses/suggestions

I order my comments in this section approximately by the order they appear in the paper.

1. In lines 34 and 37, references [41] and [29] appear at the beginning of the line, which the authors can easily avoid in latex by using the tilde character ("\~"): `text~\cite{...}`.
2. Figure 1, pane B, indicates that orthonormal bases are able to distinguish nodes that were indistinguishable in non-orthonormal bases. However, it seems to me that those points were already distinguishable before? What am I missing here? Also, do these points correspond to points in the dataset (like pane A), or are these just for illustration purposes?
3. I like the highlight in lines 84-85 (and in other places in the paper) that ABIDE is the only dataset "fully" accessible online. However, I do not see this as a big issue in practice: most of the brain datasets like ADNI, PPMI, HCP, and NACC are quite easy to be accessed, and the logistics in place are more as a general control given we are talking about sensitive human data. Most of the times there's preprocessed data available as well.
4. In section 2.1 maybe the authors will like to know about this (possibly) interesting review on GNNs: arXiv:2105.13137
5. Line 111: Typo in "STOA".
6. From figure 2/section 3.1, I can see that the authors used a symmetric weighted matrix as input. Given the authors are using the Transformer model, have the authors thought of using only the upper-half of the matrix as input to reduce model complexity? Or maybe skip this step altogether with the models I mentioned in my point (1) from major weaknesses?
7. In line 145 the "Connection Profile" is capitalised. Authors probably want it non-capitalised (as in line 139)
8.  Typo in line 187: missing "initialization" after "Xavier uniform".
9.  Line 207 has a paragraph which starts as "The second method". I believe readibility would be improved if the method is actually specified here.
10. Section 3.3.2 seems misplaced. It mentions possible generalisations of OCRead to other domains, but this is not explored in the paper. Wouldn't this be better fit in a discussion section or in conclusion?
11. RQ3, in line 244, uses the term "interpretability", but it seems to me the authors are actually looking to the term "explainability" as (for example) defined in 10.1016/j.inffus.2019.12.012. I am aware these terms might be used in different ways so I'd ask the authors to include a reference to which framework they are using.
12. In section 4.1, I think it is important to mention which tools were used, specifically DGL and pytorch from what I saw in the code (and reference accordingly).
13. Typo line 249: "Rs-fMRI" is not capitalised.
14. In line 263 and forward, is rs-fMRI also used from the ABCD dataset, or some task-based one?
15. In Table 1, why was a t-test performed (e.g., under what normality assumptions?)?
16. Given a statistical test was used in Table 1, why is there no statistical test in table 2?
17. The paper seems to provide an overly optimistic interpretation of figure 3. Sometimes variation in plot is very small (in the order of 1-2%), but the authors seem to ignore this fact? Maybe the authors could caution about this interpretation as, from what I see, a Random initialisation could be preferred for situations where model complexity can be a problem and a non-significant drop in performance is ok.
18. Typo in line 339: it is RQ3, not RQ4?
19. In Figure 4, I find it difficult to agree with the statement that orthonormal initialisations produce more discriminative values. The way I see it, I think the authors mean that visually (to the human eye) we can see some divisions on the *ordered* set of features, but that doesn't mean necessarily that the variability seen in the Random part of ABIDE is less discriminative (maybe the groups are there, not just ordered). I think this section needs to do more to be able to prove what is said.

---

> ### Author Response · Authors · 2022-08-02
> **Authors' initial rebuttal to Reviewer xfsP (part 4)**
>
> > Q11: RQ3, in line 244, uses the term "interpretability", but it seems to me the authors are actually looking to the term "explainability" as (for example) defined in 10.1016/j.inffus.2019.12.012. I am aware these terms might be used in different ways, so I'd ask the authors to include a reference to which framework they are using.
>
> A: Thanks for your insightful suggestion! We think "explainability" is more precise in our scenario and have revised it throughout the paper to make it consistent.
>
> > Q12: In section 4.1, I think it is important to mention which tools were used, specifically DGL and pytorch from what I saw in the code (and reference accordingly).
>
> A: Thanks for the helpful suggestion. Here are the packages we used along with their versions:
> pandas=1.3.4,
> torch=1.10.0,
> numpy=1.21.4,
> scipy=1.7.1,
> pyyaml=6.0,
> scikit-learn=1.0.1,
> tqdm=4.62.3.
> We are not using DGL for our model, but some baselines depend on the DGL package. We will add a list describing packages to the code repository and update section 4.1.
>
>
> > Q14: In line 263 and forward, is rs-fMRI also used from the ABCD dataset, or some task-based one?
>
> A: rs-fMRI
>
> > Q15: In Table 1, why was a t-test performed (e.g., under what normality assumptions?)?
>
> A: We add a t-test to verify the improvement since the margins between brainnetTF and baselines are not large enough for observation.
>
> > Q16: Given a statistical test was used in Table 1, why is there no statistical test in table 2?
>
> A: For table 2 we focus on observing the influence of readout functions for various Transformer architectures. Therefore the significant test is not performed between the best readout function with others.
>
> > Q17: The paper seems to provide an overly optimistic interpretation of figure 3. Sometimes variation in plot is very small (in the order of 1-2%), but the authors seem to ignore this fact? Maybe the authors could caution about this interpretation as, from what I see, a Random initialization could be preferred for situations where model complexity can be a problem and a non-significant drop in performance is ok.
>
> A: 1-2% improvement is good enough for most datasets. The time complexity of orthonormal Initialization is $\mathcal{O}(K^2V)$, where $V$ is the number of nodes and $K$ is the number of clusters. $K$ is usually less than 100. Therefore, orthonormal initialization is not a heavy burden.
>
> > Q19: In Figure 4, I find it difficult to agree with the statement that orthonormal initializations produce more discriminative values. The way I see it, I think the authors mean that visually (to the human eye) we can see some divisions on the ordered set of features, but that doesn't mean necessarily that the variability seen in the Random part of ABIDE is less discriminative (maybe the groups are there, not just ordered). I think this section needs to do more to be able to prove what is said.
>
> A: Thanks for pointing out your concerns. To further prove our observations quantitively, we calculate the difference score $d$ based on the formula $d =   \sum_{i}^{K}\sum_{j}^{V}\frac{\lvert P_{ij}^{female}-P_{ij}^{male}\rvert}{KV}$, where $V$ is the number of nodes and $K$ is the number of clusters. After running the t-test, we found the margins between random and orthonormal on both ABIDE and ABCD are significant, which is consistent with our conclusion. We will add this part to Sec 4.4 in the revision.
> |Dataset|Method|Difference score|
> | :----: | :-----: | :---------: |
> |ABIDE|Random|0.067±0.016|
> |ABIDE|Orthonormal|0.085±0.015|
> |ABCD|Random|0.125±0.010|
> |ABCD|Orthonormal|0.142±0.014|

---

> ### Author Response · Authors · 2022-08-02
> **Authors' initial rebuttal to Reviewer xfsP (part 3)**
>
> ### Other small weaknesses/suggestions
>
> We deeply thank the reviewer for these detailed suggestions on improving the presentation of our work. We have carefully fixed these mentioned typos and unclear statements in our updated PDF, and conducted more careful proofreading to find and fix other typos. For the rest of the questions, we summarize and answer them as below:
>
>
> > Q1, Q5, Q7, Q8, Q9, Q13, Q18
>
> A: Thanks for your suggestions! We have fixed this issue in our revised submission.
>
> > Q2: Figure 1, pane B, indicates that orthonormal bases are able to distinguish nodes that were indistinguishable in non-orthonormal bases. However, it seems to me that those points were already distinguishable before? What am I missing here? Also, do these points correspond to points in the dataset (like pane A), or are these just for illustration purposes?
>
> A: In the non-orthonormal bases of Figure 1(b), there still exist overlaps between two groups of nodes, making them indistinguishable from a plane, while in the orthonormal bases, those two groups can be separated completely. The figure here is just for illustration purposes. Specifically, we randomly generate two groups of nodes with different centers based on the normal distribution, then convert the bases from non-orthonormal to orthonormal.
>
> > Q3: I like the highlight in lines 84-85 (and in other places in the paper) that ABIDE is the only dataset "fully" accessible online. However, I do not see this as a big issue in practice: most of the brain datasets like ADNI, PPMI, HCP, and NACC are quite easy to be accessed, and the logistics in place are more as a general control given we are talking about sensitive human data. Most of the times there's preprocessed data available as well.
>
> A: We admit that this may not be a big issue for the neuroimaging community, but it is indeed a barrier for machine learning researchers with a bare knowledge of neuroimaging analysis. It is difficult for machine learning researchers to figure out how to access the data and what datasets can be used for neural network model development.
>
> > Q4: In section 2.1 maybe the authors will like to know about this (possibly) interesting review on GNNs: arXiv:2105.13137
>
> A: Thanks for this helpful information! We have added this review paper to our revised submission.
>
> > Q6: From figure 2/section 3.1, I can see that the authors used a symmetric weighted matrix as input. Given the authors are using the Transformer model, have the authors thought of using only the upper-half of the matrix as input to reduce model complexity? Or maybe skip this step altogether with the models I mentioned in my point (1) from major weaknesses?
>
> A: Thanks for your comment. It is indeed an interesting direction to explore. However, we cannot simply discard the bottom half of the matrix to reduce the time complexity since graph transformers or GNNs require each node's embedding to share the same dimension, and dropping the bottom half would limit the initialization information for each node.
>
> > Q10: Section 3.3.2 seems misplaced. It mentions possible generalisations of OCRead to other domains, but this is not explored in the paper. Wouldn't this be better fit in a discussion section or in conclusion?
>
> A: Thanks for your suggestion! We think this makes sense and will move this section to the discussion section in the revised paper.

---

> ### Author Response · Authors · 2022-08-02
> **Authors' initial rebuttal to Reviewer xfsP (part 2)**
>
> > Q3: I believe the authors are mistakenly using "gender" when they mean biological sex. As far as I know, these datasets do not provide information on the gender of the people in the datasets, but rather the biological binary sex. I therefore request the authors to correct this for example in figure 2 and in the other occurrences of "gender".
>
> A: Thanks for your great suggestion. We will consistently modify this language everywhere in the revision.
>
> > Q4: I do not agree with the paper's justification in lines 270-272 that AUROC is the most proper performance metric, given for example that in the medical context a model needs to use a single threshold to decide on the positive/negative output, and AUROC tests all thresholds in practice. Besides, the model is trained such that the highest AUROC performance on the validation set is attained, which could artificially inflate these results and we do not know how they affect other very important metrics in the medical context like sensitivity, specificity, or even model calibration. I request the authors to at least add values on sensitivity and specificity (or equivalent) to show the strengths of this work.
>
> A: This is really a valuable suggestion. We have added two more metrics, sensitivity and specificity, as shown in response to Q1 and will update Table 1 correspondingly in the revision.
>
>
> >Q5: In Table 1 I find it difficult to understand how fair the comparisons are. Are these models similar in terms of number of layers? Or number of parameters? Given BrainNetTF seems to have had some sort of hyperparameter search, it is probably unfair to compare with out-of-the-box models?
>
> A: Sure, we calculate the number of parameters for each model and show the results below. It is shown that our model is larger than GNN and CNN models but has a similar size to other transformer models.
>
> |Dataset|Method|#Para|
> | :----: | :-----: | :---------: |
> |ABIDE|BrainNetTF|20.2M|
> |ABIDE|BrainNetCNN|0.93M|
> |ABIDE|FBNetGen|0.55M|
> |ABIDE|VanillaTF|15.6M|
> |ABIDE|SAN|57.7M|
> |ABIDE|Graphormer|1.23M|
> |ABIDE|BrainGB|1.08M|
> | :----: | :-----: | :---------: |
> |ABCD|BrainNetTF|45.1M|
> |ABCD|BrainNetCNN|0.93M|
> |ABCD|FBNetGen|1.18M|
> |ABCD|VanillaTF|32.7M|
> |ABCD|SAN|186.7M|
> |ABCD|Graphormer|1.66M|
> |ABCD|BrainGB|1.49M|
>
> As for the hyperparameter tuning, the comparison is fair since grid search is applied for hyper-parameter tuning for our model as well as important parameters in all other baselines. To be specific, for BrainGB, the readout function is searched from {mean, max, concat} and the message-passing function is searched from {Edge weighted, Node edge concat, Node concat}. For BrainGNN, the learning rate is searched in {0.01, 0.005, 0.001} and the feature dimension is searched in {100, 200}. For FBNetGen, different encoders {1D-CNN, GRU} are tested with different hidden dimensions {8, 12, 16}. For BrainNetCNN, dropout rate is selected from {0.3, 0.5, 0.7}. For VanillaTF, the number of transformer layers is searched from {1, 2, 3} with the number of headers from {2, 4, 6}. For SAN, we test LPE hidden dimension from {4, 8, 16}, the number of LPE and GT transformer layers from {1, 2}, and the number of headers from {2, 4} with 50 epochs of training. For Graphormer, the number of encoder layers is selected from {1, 2} and the embed dimension is from {256, 512}.

---

> ### Author Response · Authors · 2022-08-02
> **Authors' initial rebuttal to Reviewer xfsP (part 1)**
>
> We thank the reviewer for the detailed comments and suggestions that help us improve this work's quality. Following are our detailed responses to the reviewer's questions.
>
> > Before going into it, I must ask the authors to please double-check their terms to anonymise in the code shared because while I was analysing the code, I found at least one .yaml file under the setting folder with an absolute path.
>
> Thanks for the reminder. We have cleared these YAML files.
>
> ### Major weaknesses
>
> > Q1: The paper cites Kan et al. [25] to justify the idea of having the network to learn the graph structure, and uses the model presented in this reference (FBNETGEN) as another baseline for comparison. Although I know that choosing baselines can be a very subjective topic, I must say that selecting only one baseline for a learnable brain network is not enough. I have seen at least two possible papers learning a graph structure (arXiv:2002.04999, 10.1016/j.media.2020.101709) which were never evaluated on ABIDE/ABCD and could therefore be another interesting contribution of this paper to use this previous literature on unexplored datasets, even though the 2 papers I leave here had some analysis on medical images and were not used in the original FBNETGEN paper. From Table 1, I noticed that BrainGB and FBNETGNN seem to come from the same group as they come from the two references used to motivate the some choices on the model (refs 12 and 25), which is another non-subjective reason to include another baseline from another paper. Finally, the "learnable brain network" type in Table 1 is the one with the closest performance to BrainNetTF and therefore another baseline of the same type would improve the significance of this model. For all these reasons, I believe the authors need to add at least another meaningful baseline on the "learnable brain network" category in order for their results to be more sound.
>
> A:  Thanks for your suggestion! Following your and other reviewers' suggestion, we have added three baselines of BrainGNN [1], BrainNetGNN [2] and DGM [3], with performances summarized in the updated Table 1 in our first comment for all reviewers above. BrainGNN is the baseline for fixed brain networks, while BrainNetGNN and DGM are baselines for learnable brain networks.
>
> > Q2: On lines 170-171 the authors claim that "none of the existing readout functions leverages the properties of brain networks that nodes in the same functional modules tend to have similar behaviors and clustered representations". After reading this sentence I immediately thought about the DiffPool paper (arXiv:1806.08804) and so many other papers adapted from this differentiable graph pooling module which, in practice, forms different clusters as a way to pool the nodes representations. I know of at least one recent paper (10.1016/j.media.2022.102471) using it in the medical context as readout function, so I believe the authors should probably downtown this sentence, and find a more complex (and who knows fairer) readout/pooling operator to compare OCRead with. From what I see in the code, the authors used DGL with pytorch, which I believe has implementations available for more complex pooling layers (https://docs.dgl.ai/en/0.8.x/api/python/nn-pytorch.html#global-pooling-layers) or DiffPool (https://github.com/dmlc/dgl/blob/master/examples/pytorch/diffpool/model/tensorized_layers/diffpool.py).
>
> A: In our original paper, we compared with SortPooling, which is the pooling function used in BrainGNN. Following your suggestion, here we add an additional comparison with DiffPool, with the results shown below. Unfortunately, DiffPool gives pretty poor results, and we believe this is because this method does not leverage specific properties of brain networks, such as the fixed order and number of nodes.
>
> | Dataset |     Readout     |    SAN|Graphormer|VanillaTF|
> |:-------:|:--------------:|:---------:|:---------:|:-----------:|
> |ABIDE|MEAN  | 63.7±2.4 | 50.1±1.1| 73.4±1.4|
> |ABIDE|MAX  | 61.9±2.5 | 54.5±3.6| 75.6±1.4|
> |ABIDE|SUM  | 62.0±2.3 | 54.1±1.3| 70.3±1.6|
> |ABIDE|SortPooling |68.7±2.3 | 51.3±2.2| 72.4±1.3  |
> |ABIDE|DiffPool | 57.4±5.2 |50.5±4.7| 62.9±7.3  |
> |ABIDE|CONCAT | 71.3±2.1 |63.5±3.7| 76.4±1.2  |
> |ABIDE|OCRead| 70.6±2.4 |64.9±2.7 | 80.2±1.0 |
> |:-------:|:--------------:|:---------:|:---------:|:-----------:|
> |ABCD|MEAN   |88.5±0.9 | 87.6±1.3|91.3±0.7 |
> |ABCD|MAX    | 87.4±1.1 | 81.6±0.8| 94.4±0.6|
> |ABCD|SUM    | 84.2±0.8 | 71.5±0.9| 91.6±0.6|
> |ABCD|SortPooling  | 84.6±1.1 | 86.7±1.0| 89.9±0.6|
> |ABCD|DiffPool  | 78.1±1.5 | 70.0±1.9| 83.9±1.3 |
> |ABCD|CONCAT | 90.1±1.2 | 89.0±1.4| 94.3±0.7 |
> |ABCD|OCRead  | 91.2±0.7 | 90.2±0.7| 96.2±0.4|

---

> ### Comment · Reviewer_xfsP · 2022-08-03
> **My answer to the authors' rebuttal**
>
> I thank the authors for the clarifications on this rebuttal - I am happy with how the authors tackled my major concerns. I have questions on the rebuttal to other reviewers, but for those ones I've commented at each separately.
>
> Overall, I believe this improved the quality of the work. However, it seems clear to me that some limitations of the work cannot be improved during this rebuttal process, for instance: (1) try another readout function, for better baselines comparison, beyond diffpool which showed some limitations in previous works already (eg 10.1016/j.media.2022.102471) and given the authors comment to my question 2 that this readout function does not leverage some properties of brain networks, (2) trying more homogeneous datasets (eg. HCP, which although I appreciate that for the ML community might not be the easiest thing to do, it has been around for a while and should be possible to access in an applied neuroscientific work at a prestigious conference like neurips), (3) trying some other predictive variables, which maybe could be easier found in other datasets (connected to reviewer wuUg's Q1), and (4) the fact that this is yet another paper on another transformer adaption with limited improvements to previous models, and thus doesn't make it groundbreaking impact (even if with good methods and significant contribution to neurips). I present these limitations as a way to justify why I cannot give a very high score (i.e., close to 10); however, the improvements are more than enough to increase my original scores. I'd like to highlight that I appreciate the fact that the authors did some more systematic hyperparameter sweep across the different architectures, despite the limitations of grid search on a limited set of variables (compared for instance to random search, as one can see in Bergstra and Bengio's "Random search for hyper-parameter optimization").
>
> Before deciding on the specific changes to my scores, I just have one more question to ask (beyond the ones I'm leaving to other reviewers): the authors did not tackle my "Limitations" section of my review. Are they going to add some sentences on this?

---

> > ### Author Response · Authors · 2022-08-03
> > **Many thanks and further answers (part 2)**
> >
> > >  The paper contains predictions using human brain data, with some labels being sex and diagnosis. Therefore, I find it incorrect that the paper does not follow the recommendations on writing about societal impact when the prediction of these labels in specific can have so many different impacts in society. I believe the acknowledgement of these variables would be the bare minimum needed for a NeurIPS paper (maybe starting from the last sentence of conclusion). With regards to limitations, I believe they were tackled properly in Conclusion and in other sections of the paper. However, as a consequence of the answers to my questions this will be be better after the rebuttal process.
> >
> > A: Thank you for the comment. We will include the following discussions of the societal impact of our work in the revised version:  *Neuroimaging technologies, including functional magnetic resonance imaging (fMRI) are powerful noninvasive tools for examining the brain functioning. There is an emerging nation-wide interest in conducting neuroimaging studies for investigating the connection between the biology of the brain, and demographic variables and clinical outcomes such as mental disorders. Such studies provide an unprecedented opportunity for cross-cutting investigations that may offer new insights to the differences in brain function and organization across subpopulations in the society (such as biological sex and age groups) as well as reveal neurophysiological mechanisms underlying brain disorders (such as psychiatric illnesses and neurodegenerative diseases). These studies have a tremendous impact in social studies and biomedical sciences.  For example, mental disorders are the leading cause of disability in the USA and roughly 1 in 17 have a seriously debilitating mental illness. To address this burden, national institutions such as NIH have included brain-behavior research as one of their strategic objectives and stated that sound efforts must be made to redefine mental disorders into dimensions or components of observable behaviors that are more closely aligned with the biology of the brain. Using brain imaging data to predict diagnosis has great potential to result in mechanisms that target for more effective preemption and treatment.*

---

> > ### Author Response · Authors · 2022-08-03
> > **Many thanks and further answers (part 1)**
> >
> > > I thank the authors for the clarifications on this rebuttal - I am happy with how the authors tackled my major concerns. I have questions on the rebuttal to other reviewers, but for those ones I've commented at each separately.
> >
> > A: Thank you so much for leading a stringent review discussion!
> >
> > > Overall, I believe this improved the quality of the work. However, it seems clear to me that some limitations of the work cannot be improved during this rebuttal process, for instance: (1) try another readout function, for better baselines comparison, beyond diffpool which showed some limitations in previous works already (eg 10.1016/j.media.2022.102471) and given the authors comment to my question 2 that this readout function does not leverage some properties of brain networks, (2) trying more homogeneous datasets (eg. HCP, which although I appreciate that for the ML community might not be the easiest thing to do, it has been around for a while and should be possible to access in an applied neuroscientific work at a prestigious conference like neurips), (3) trying some other predictive variables, which maybe could be easier found in other datasets (connected to reviewer wuUg's Q1), and (4) the fact that this is yet another paper on another transformer adaption with limited improvements to previous models, and thus doesn't make it groundbreaking impact (even if with good methods and significant contribution to neurips). I present these limitations as a way to justify why I cannot give a very high score (i.e., close to 10); however, the improvements are more than enough to increase my original scores. I'd like to highlight that I appreciate the fact that the authors did some more systematic hyperparameter sweep across the different architectures, despite the limitations of grid search on a limited set of variables (compared for instance to random search, as one can see in Bergstra and Bengio's "Random search for hyper-parameter optimization").
> >
> > A: Again we really appreciate the detailed summarizations. We agree with all these points. Here are just some of our reflections, without trying to further influence your judgements. (1) There are indeed many other readout functions such as topkPooling, Edgepooling, SAG pooling and GlobalAttention suggested by reviewer CYN4– topkPooling is actually similar to the sortPool we compared, and SAG pooling is similar to DiffPool. With sortPool and DiffPool, we think typical existing readout functions have been covered (at least conceptually). We are happy to further include results on other readout functions, but our key argument that none of them was designed to leverage brain network properties is already established. (2) HCP has indeed been there for a while, but it is unfortunately not yet publicly available with ready-to-use *brain network* data, which makes it not only challenging but also unnecessary for us to include the results on HCP (challenging because we don’t have access to the data, and unnecessary because everyone using that data preprocessed it differently). (3) As we explained to reviewer wuUg, biological sex prediction is in fact a meaningful task, and almost the only task currently with enough labels to conduct meaningful experiments on ML methods. (4) We won’t argue this. While not every paper is “goundbreaking” (thus getting a score close to 10), we believe our work is at least bringing substantial value to both the graph mining and neuroscience communities.
> >
> > > Before deciding on the specific changes to my scores, I just have one more question to ask (beyond the ones I'm leaving to other reviewers): the authors did not tackle my "Limitations" section of my review. Are they going to add some sentences on this?
> >
> > A: Sorry for missing the question in the Limitation section. Please see our answer in the second part below.

---

> ### Comment · Reviewer_xfsP · 2022-08-07
> **Change on my scores**
>
> In my opinion, the authors tackled my question in a satisfactory way, and I am making these changes to my review:
>
> Soundness: 2 > 3
> Contribution: 2 > 3
> Rating: 5 > 7

---

> > ### Author Response · Authors · 2022-08-08
> > **Thank you for the really responsible reviews and feedback**
> >
> > Thank you reviewer xfsP. It was our great fortune to receive reviews as detailed, insightful, and constructive as yours.

---

### Official Review · Reviewer_wuUg · 2022-07-11

**Rating:** 6
**Confidence:** 4
**Soundness:** 3 good
**Presentation:** 3 good
**Contribution:** 3 good

**Summary:**

The paper presents a graph transformer for deriving effective representations of brain networks. In particular, this handles several caveats of brain networks: 1) use of connection profiles as positional embeddings and 2) deriving per-graph representation by projecting the node embeddings to orthonormal centroids for initialization. The downstream tasks using these embeddings on two public fMRI datasets show superior performance on autism spectrum disorder and gender prediction.

**Questions:**

The paper is overall solid. There are some questions related to the weaknesses section:
1. Structural connectivities are often not fully connected via heuristics (e.g., thresholding) and/or biology (e.g., no fiber bundles between certain regions). Would the proposed method also properly leverage the connectivity profile?
2. Was BrainNetTF tested with other readout functions?
3. Why was gender chosen to be predicted using the ABCD dataset? Were there no other meaningful response variables to study? I am not sure how meaningful gender prediction is in general.

[Update after the rebuttal]
I am keeping my score (6). Most of my concerns were addressed, so I appreciate the authors' effort. I still think the stratified sampling of the sites is still just a simple heuristic.

**Limitations:**

Yes

**Strengths And Weaknesses:**

Strengths:
+ The paper is overall well-written and easy to understand.
+ The experiments are thorough with numerous baselines and ablation studies.
+ The theoretical justifications for the benefits of orthonormality in a statistical manner are interesting.

Weaknesses:
- Predicting the gender with the ABCD dataset is not particularly interesting or useful.
- If the soft clustering based on the proposed orthonormal bases shows empirical benefits, it would have been interesting to also see how the clusters imply additional neuroscientific findings based on their regions or functional modules.
- The stratified splitting based on the site information is a common procedure.

---

> ### Author Response · Authors · 2022-08-02
> **Authors' initial rebuttal to Reviewer wuUg (part 2)**
>
> > Q4: Structural connectivities are often not fully connected via heuristics (e.g., thresholding) and/or biology (e.g., no fiber bundles between certain regions). Would the proposed method also properly leverage the connectivity profile?
>
> A: Yes. Our proposed BrainNetTF does not require the input graph to be a complete graph. In section 3.3.2, we have discussed the potential usage of our method for structural connectivities.
>
> > Q5: Was BrainNetTF tested with other readout functions?
>
> A: Yes, we have equipped BrainNetTF with other readout functions, such as MEAN, MAX, SUM, SortPooling, and Concat. The results can be found in the VanillaTF column of Table 2. Since MHSA+Concat=VanillaTF and MHSA+OCRead=BrainNetTF, testing VanillaTF with different readout functions is equal to testing BrainNetTF with different readout functions. Besides, we have also added experiments for an additional readout function where the VanillaTF is equipped with DiffPool [1]. The results are summarized in the following table.
>
> | Dataset | Readout | VanillaTF|
> |:-------:|:--------------:|:---------:|
> |ABIDE|MEAN | 73.4±1.4|
> |ABIDE|MAX | 75.6±1.4|
> |ABIDE|SUM | 70.3±1.6|
> |ABIDE|SortPooling| 72.4±1.3 |
> |ABIDE|DiffPool| 62.9±7.3 |
> |ABIDE|CONCAT| 76.4±1.2 |
> |ABIDE|OCRead | 80.2±1.0 |
> |:-------:|:--------------:|:---------:|
> |ABCD|MEAN |91.3±0.7 |
> |ABCD|MAX | 94.4±0.6|
> |ABCD|SUM | 91.6±0.6|
> |ABCD|OCRead | 89.9±0.6|
> |ABCD|DiffPool | 83.9±1.3 |
> |ABCD|CONCAT | 94.3±0.7 |
> |ABCD|OCRead | 96.2±0.4|
>
>
> [1] Ying, Zhitao, et al. "Hierarchical graph representation learning with differentiable pooling." Advances in neural information processing systems 31 (2018).

---

> ### Author Response · Authors · 2022-08-02
> **Authors' initial rebuttal to Reviewer wuUg (part 1)**
>
> > Q1: Predicting the gender with the ABCD dataset is not particularly interesting or useful. (Why was gender chosen to be predicted using the ABCD dataset? Were there no other meaningful response variables to study? I am not sure how meaningful gender prediction is in general.)
>
> A: ABCD does not aim to study a particular disease. It is a large-scale dataset aiming to study the behavioral and brain development of Adolescents, which is a longitudinal study starting at the ages of 9-10 and following participants for 10 years. Since sexuality is an important aspect of adolescent development, biological sex prediction is a critical and meaningful task for ABCD. Many papers [1, 2, 3, 4] have focused on this task using brain networks.
>
> Besides, the purpose of this paper is to propose a generic transformer-based model for brain networks. Hence, we need to use a task that is not only related to brain networks but also easy to evaluate. Although ABCD provides thousands of labels for social, emotional, and cognitive development, mental health, and substance use, most of these labels contain too many missing values. Here is part of our analysis from thousands of tasks after removing labels with too many missing values. These tasks, while maybe “more interesting” to study, simply lack sufficient and balanced labels for the comparison of different models.
>
> | Label Description | Type | Label Distribution |
> | :----------------------------------------------- | :------: | :---------------------------------------- |
> | Diagnosis - Hallucinogen Use Disorder Present | Discrete | Label 1: 5904 (99.97%), Label 2: 2 (0.03%) |
> | Diagnosis - Tobacco Use Disorder Present | Discrete | Label 1: 5898 (99.86%), Label 2: 8 (0.14%) |
> | Diagnosis - Cannabis Use Disorder Present | Discrete | Label 1: 5904 (99.97%), Label 2: 2 (0.03%) |
> | Diagnosis - Alcohol Use Disorder Present | Discrete | Label 1: 5905 (99.98%), Label 2: 1 (0.02%) |
> | Diagnosis - Schizophrenia (F20.9) | Discrete | Label 1: 5903 (99.97%), Label 2: 2 (0.03%) |
> | Diagnosis - Bulimia Nervosa (F50.2) PRESENT | Discrete | Label 1: 5901 (99.97%), Label 2: 2 (0.03%) |
> | Diagnosis - Binge-Eating Disorder (F50.8) CURRENT | Discrete | Label 1: 5858 (99.24%), Label 2: 45 (0.76%) |
>
> [1] Alexandra S. Potter et al. Measurement of gender and sexuality in the Adolescent Brain Cognitive Development (ABCD) study, Developmental Cognitive Neuroscience,Volume 53, 1878-9293 (2022).
> [2] Gennatas, E. D. et al. Age-Related Effects and Sex Differences in Gray Matter Density, Volume, Mass, and Cortical Thickness from Childhood to Young Adulthood. J Neurosci 37, 5065–5073 (2017).
> [3] Gur, R. E. & Gur, R. C. Sex differences in brain and behavior in adolescence: Findings from the Philadelphia Neurodevelopmental Cohort. Neurosci Biobehav Rev 70, 159–170 (2016).
> [4] Satterthwaite, T. D. et al., Linked Sex Differences in Cognition and Functional Connectivity in Youth. Cerebral Cortex, 25, 2383–2394
>
> > Q2: If the soft clustering based on the proposed orthonormal bases shows empirical benefits, it would have been interesting to also see how the clusters imply additional neuroscientific findings based on their regions or functional modules.
>
> A: This is a great suggestion. In each attention layer of a transformer-based model, every node updates its own embedding by the weighted sum of all other nodes’ embeddings. Therefore, the embedding of a node no longer corresponds to a physical brain region after the first attention layer. Thus, only the weights of the first attention layer are directly meaningful for region or module analysis. We indeed illustrated this in Figure 4 in our original draft, which shows a matching pattern with the ground-truth functional modules. Besides, with the current method, it is meaningless to find the soft clusters of the learned node embeddings since these embeddings have been through several attention layers, but we do find the optimal cluster numbers to be somewhat consistent with the numbers of ground-truth clusters, as showing in Figure 3 (a), indicating a potential to further decouple the attentions and find soft clusters of physical brain regions in the future.
>
> > Q3: The stratified splitting based on the site information is a common procedure.
>
> A: This is correct in principle, but there has not been an effort to standardize the stratified splitting process for ABIDE. Our contribution here is to provide this standard so as to really conduct fair/meaningful performance comparisons and facilitate future research.

---

> ### Author Response · Authors · 2022-08-08
> **Thanks for acknowledging our contributions**
>
> Dear reviewer wuUg,
>
> Many thanks for your initial positive acknowledgments of our contributions and for holding on to them after the rebuttal.
>
> Best,
> Authors

---

### Official Review · Reviewer_CYN4 · 2022-07-11

**Rating:** 3
**Confidence:** 5
**Soundness:** 2 fair
**Presentation:** 2 fair
**Contribution:** 1 poor

**Summary:**

In this work, the authors have proposed a self-attention based graph model to classify a) ASD and b) gender using functional neuroimaging fMRI datasets. The authors also proposed a new readout method based on clustering for graph classification. The proposed method seems to perform a bit better in the classification of both tasks in comparison to other baseline methods.


**Questions:**

- Figure 2 shows multiple $X$ matrices and it is unclear how they were acquired. My guess is it's the same matrix, rather the self-attention module receiving rows of the matrix.
- How is the matrix $X$ computed?
- If classification performance is the goal then why not use simple ML models like SVM, LR on FC matrices computed via PCC. With hand-crafted features (FC matrices), ML models give better classification performance.
- How were the results for comparing studies calculated?
- Why not use HCP data for gender classification as it is publically available?

**Limitations:**

- Results were not compared against many other existing relevant studies. Mentioned in weaknesses.
- The readout method compared against basic methods such as sum, max, etc. but not against serious and widely available contenders.
- Marginal improvement in classification accuracy ($\approx1$%).
- No cross-validation, no multiple test folds.
- Missing details about input data ($X$).


**Strengths And Weaknesses:**

## Strengths
- The biggest strength is the classification performance of the model for ASD and the gender classification tasks. The classification performance is ~1% better in terms of accuracy when comparing with next best (FBNETGNN)

## Weaknesses
- There are some comments in the paper which are either incorrect or not properly justified. For example, lines 43-44 mention "First, a brain network is a correlation matrix defined on a complete graph."  This is not correct as correlation matric computed using methods like Pearson’s Correlation Cofficient is a way to measure brain connectivity. There can be many other ways to represent brain network e.g. Effective Connectivity, Directed connectivity, Entropy based, causality based (Granger causality), etc. Lines 49-51: it would be better to explain how a 1D time-series of a voxel has positional information.
- The authors do not mention how they compute the $X \in RV *V$ matrix.
- The authors cite multiple self-attention and graph based methods but do not compare with them, for example BrainGNN, STAGIN. It would be better to compare with these methods and others like these on the same task.
- There are others self-attention and graph based methods developed for neuroimaging datasets.
   1. A Deep Learning Model for Data-Driven Discovery of Functional Connectivity https://doi.org/10.3390/a14030075
   2. Deep Dynamic Effective Connectivity Estimation from Multivariate Time Series https://arxiv.org/abs/2202.02393
- It is unclear how the results used for comparison of the methods are acquired. Are they taken from other papers? Was the code open source? How the hyper-parameters were tuned for these methods?
- There are many readout methods developed already which perform much better than the simple ones (add, max, avg, min) such as attention based, topkPooling, Edgepooling, SAG pooling and many more. These methods are also acquire a smaller graph as the readout method proposed by the authors. Other global pooling methods also exist such as GlobalAttention. Comparison with them is lacking.
- Similar concerns arise regarding the results in Table 2 as well. The manuscript compares against basic methods while many sophisticated and more powerful methods are already proposed. Table 2 also shows that the gains by the readout method proposed by the authors are minute when comparing against CONCAT method which is still not the most sophisticated among the available.
- The authors mention that they randomly selected $70$% data for training, $10$% for val and $20$% for testing. As I understand, there is no cross-validation or testing on multiple folds. ABIDE dataset has a high variation of performance across subjects and the single-point results cannot be fully trusted. This can be a reason that in Fig 5, it is seen that the proposed model performs better on test data than val data. It is very much possible that the authors considered test subjects that generally provide high classification performance.
- The paper provide very limited interpretation of the results.

---

> ### Author Response · Authors · 2022-08-02
> **Authors' initial rebuttal to Reviewer CYN4 (part 3)**
>
> > Q7: The authors mention that they randomly selected 70% data for training, 10% for val and 20% for testing. As I understand, there is no cross-validation or testing on multiple folds. ABIDE dataset has a high variation of performance across subjects and the single-point results cannot be fully trusted. This can be a reason that in Fig 5, it is seen that the proposed model performs better on test data than val data. It is very much possible that the authors considered test subjects that generally provide high classification performance.
>
> A: We repeat this random process 5 times and show the final result (thus the reported STD). ABIDE is a dataset containing more than 1000 samples, randomly split is good enough to evaluate model performance. It is unethical to pick test subjects with higher performance, and this is not a part of our experiment. We are sorry for any missing details that have caused your misunderstanding and will add clarifications in the revised paper.
>
> > Q8: The paper provide very limited interpretation of the results.
>
> A: We are sorry but this is a very vague claim. We believe in having provided rich textual interpretation of our results in EVERY subsection of the experiments (Sec 4.2, 4.3, 4.4 to be specific). We have no idea what you are referring to as “very limited interpretation of the results” here, but we would certainly be glad to discuss more if you could kindly clarify this question.
>
> > Q9: If classification performance is the goal then why not use simple ML models like SVM, LR on FC matrices computed via PCC. With hand-crafted features (FC matrices), ML models give better classification performance.
>
> A: Classification performance is not the only goal in our work, as can be observed from our interpretation results such as in Sec 4.4. However, following your advice, we have also included simple ML models such as logistic regression and SVM with the best hyper-parameters obtained through grid search on our two datasets ([Code](https://anonymous.4open.science/r/BrainTransformer/baselines/lr_svm_baseline.py)). From the table below, we can see in both ABIDE and ABCD, that simple ML models like SVM and LR on FC matrices computed via PCC do not give better classification performance as assumed.
>
> Besides, exploring neural network models is meaningful since many SOTA technologies can be applied, like transfer learning or meta learning, which is meaningful for situations like lacking samples and labels in brain network analysis.
>
> | Method | Dataset |     AUC     |     ACC      |
> | :----: | :-----: | :---------: | :----------: |
> |   LR   |  ABIDE  | 75.97±1.09 | 68.52±1.11  |
> |  SVM   |  ABIDE  | 75.07±4.15 | 69.90±3.11  |
> |   LR   |  ABCD   | 94.06±0.26 | 87.057±0.36 |
> |  SVM   |  ABCD   | 93.52±0.54 | 86.98±0.43 |
>
>
> > Q10: Why not use HCP data for gender classification as it is publically available?
>
> We currently have not obtained access to the HCP dataset since it is restrictively available, and the generation of functional brain networks from raw brain imaging data takes significant effort. We are interested in conducting more experiments if the datasets become available, but we don’t think lacking experimental results on a specific dataset is a major drawback since in principle no paper can simply include experiments on every possible dataset.

---

> > ### Comment · Reviewer_xfsP · 2022-08-03
> > **Further question on Q17 and Q10**
> >
> > Q7. What do the authors mean by "randomly split" on ABIDE, given they mention a stratified approach in their paper?
> >
> >
> > Q10. I am a bit confused with the authors claim that HCP is restrictively available. Last time I tried to access it I don't think it was much more difficult than just creating an account to download data - has something changed in terms of access? Of course generation of functional brain networks are very difficult and require expert knowledge, but HCP provides preprocessed data with their own atlas, for instance: https://www.humanconnectome.org/study/hcp-young-adult/document/1200-subjects-data-release
> > I'd understand if the authors wanted to have the same preprocessed pipeline across all datasets used in this work, but the authors already used different atlas for ABIDE and ABCD, right? I know this is not a major drawback of this work, but it still one which I think it's important to clarify.

---

> > > ### Author Response · Authors · 2022-08-03
> > > **Answers to the further questions on Q7 and Q10**
> > >
> > > We thank the reviewer for the additional clarification questions!
> > >
> > > > Q7. What do the authors mean by "randomly split" on ABIDE, given they mention a stratified approach in their paper?
> > >
> > > A: Our stratified approach includes two steps: divide the dataset according to sites, and then conduct random data split inside each “strata (site/group)”. This is also known as “stratified random sampling”, and we will clarify this in the revised version.
> > >
> > > > Q10. I am a bit confused with the authors claim that HCP is restrictively available. Last time I tried to access it I don't think it was much more difficult than just creating an account to download data - has something changed in terms of access? Of course generation of functional brain networks are very difficult and require expert knowledge, but HCP provides preprocessed data with their own atlas, for instance: https://www.humanconnectome.org/study/hcp-young-adult/document/1200-subjects-data-release I'd understand if the authors wanted to have the same preprocessed pipeline across all datasets used in this work, but the authors already used different atlas for ABIDE and ABCD, right? I know this is not a major drawback of this work, but it still one which I think it's important to clarify.
> > >
> > > A: Thanks for your suggestions and clarifications. As far as we know, there exists no publicly available ready-to-use brain network dataset preprocessed from HCP. Yes, some preprocessed data have been released for HCP such as the one you referenced, but these are still brain imaging data and there are still many steps to be done to get the brain network data from there (and unfortunately these steps are performed rather differently across labs). Due to these reasons, we hope it is understandable that we have not included HCP in the current experiments, because (1) we do not yet have access to a readily usable brain network dataset of HCP, and (2) HCP is not yet a standard dataset for comparing brain network analysis methods (because every group who used the dataset preprocessed the dataset rather differently by themselves). This is still unfortunate, but this is the challenge of the whole community, and we are making an effort in this work by standardizing the comparison on ABIDE, which as far as we know, is the only existing dataset with publicly available preprocessed data of brain networks. Beyond this work, we are very interested in obtaining more ready-to-use brain network datasets and enabling proper sharing of them (e.g., if we cannot directly share the datasets, we can at least share some standardized preprocessing pipelines).

---

> ### Author Response · Authors · 2022-08-02
> **Authors' initial rebuttal to Reviewer CYN4 (part 2)**
>
> > Q5: It is unclear how the results used for comparison of the methods are acquired. Are they taken from other papers? Was the code open source? How the hyper-parameters were tuned for these methods? (How were the results for comparing studies calculated?)
>
> A: This is a great suggestion. We will add one subsection to our Appendix to clarify the details of compared algorithms. For [BrainGB](https://github.com/HennyJie/BrainGB), [BrainGNN](https://github.com/xxlya/BrainGNN_Pytorch), [FBNetGen](https://github.com/Wayfear/FBNETGEN), we use the author's open-source code accessible via the hyperlinks. For [SAN](https://github.com/DevinKreuzer/SAN) and [Graphormer](https://github.com/microsoft/Graphormer), we folk their repositories and modified them for the brain network datasets. For BrainNetCNN and VanillaTF, we implement them by ourselves.
> We use grid search for important hyper-parameters for these baselines based on the provided best setting. To be specific, for BrainGB, we search different readout functions {mean, max, concat} with different message-passing functions {Edge weighted, Node edge concat, Node concat}. For BrainGNN, we search different learning rates {0.01, 0.005, 0.001} with different feature dimensions {100, 200}. For FBNetGen, we search different encoders {1D-CNN, GRU} with different hidden dimensions {8, 12, 16}. For BrainNetCNN, we search different dropout rates {0.3, 0.5, 0.7}. For VanillaTF, we search the number of transformer layers {1, 2, 3} with the number of headers {2, 4, 6}. For SAN, we test LPE hidden dimensions {4, 8, 16}, the number of LPE and GT transformer layers  {1, 2} and the number of headers {2, 4} with 50 epochs training. For Graphormer, we test encoder layers {1, 2} and embed dimensions {256, 512}.
>
> > Q6: There are many readout methods developed already which perform much better than the simple ones (add, max, avg, min) such as attention based, topkPooling, Edgepooling, SAG pooling and many more. These methods are also acquire a smaller graph as the readout method proposed by the authors. Other global pooling methods also exist such as GlobalAttention. Comparison with them is lacking. Similar concerns arise regarding the results in Table 2 as well. The manuscript compares against basic methods while many sophisticated and more powerful methods are already proposed. Table 2 also shows that the gains by the readout method proposed by the authors are minute when comparing against CONCAT method which is still not the most sophisticated among the available.
>
> A: The more complex readout functions "topkPooling, Edgepooling, SAG pooling" do not necessarily mean better performance, especially since they are not specifically designed for brain network special domain (CONCAT is not a "complex" readout function here but rather a specific design for brain networks– one can use CONCAT only for networks with a fixed number and order of nodes such as brain networks).
> For the more complex readout functions, we followed the advice to add the popular pooling method of DiffPool [1] as an example below, and it does not lead to better performance. Given the limited time, we could not finish experiments with other more complex readout functions, but we will add the results gradually as they become available and include them in the revised paper.
>
> [1] Ying, Zhitao, et al. "Hierarchical graph representation learning with differentiable pooling." Advances in neural information processing systems 31 (2018).
>
> | Dataset |     Readout     |    SAN|Graphormer|VanillaTF|
> |:-------:|:--------------:|:---------:|:---------:|:-----------:|
> |ABIDE|MEAN  | 63.7±2.4 | 50.1±1.1| 73.4±1.4|
> |ABIDE|MAX  | 61.9±2.5 | 54.5±3.6| 75.6±1.4|
> |ABIDE|SUM  | 62.0±2.3 | 54.1±1.3| 70.3±1.6|
> |ABIDE|SortPooling |68.7±2.3 | 51.3±2.2| 72.4±1.3  |
> |ABIDE|DiffPool | 57.4±5.2 |50.5±4.7| 62.9±7.3  |
> |ABIDE|CONCAT | 71.3±2.1 |63.5±3.7| 76.4±1.2  |
> |ABIDE|OCRead| 70.6±2.4 |64.9±2.7 | 80.2±1.0 |
> |ABCD|MEAN   |88.5±0.9 | 87.6±1.3|91.3±0.7 |
> |ABCD|MAX    | 87.4±1.1 | 81.6±0.8| 94.4±0.6|
> |ABCD|SUM    | 84.2±0.8 | 71.5±0.9| 91.6±0.6|
> |ABCD|OCRead  | 84.6±1.1 | 86.7±1.0| 89.9±0.6|
> |ABCD|DiffPool  | 78.1±1.5 | 70.0±1.9| 83.9±1.3 |
> |ABCD|CONCAT | 90.1±1.2 | 89.0±1.4| 94.3±0.7 |
> |ABCD|OCRead  | 91.2±0.7 | 90.2±0.7| 96.2±0.4|

---

> ### Author Response · Authors · 2022-08-02
> **Authors' initial rebuttal to Reviewer CYN4 (part 1)**
>
> We appreciate the time and detailed comments of the reviewer. However, we believe most of the reviewer’s concerns are around the experimental details, and many of these are really “misunderstandings” (e.g., Q3, Q7, Q9), “minor/easily fixed” (e.g., Q1, Q2, Q4), while Q8 is pretty vague. While we believe these points should not sum up into an overall recommendation of “reject”, we are very grateful to have the opportunities to improve our work, and hope the reviewer could reconsider the recommendation as well.
>
> > Q1: There are some comments in the paper which are either incorrect or not properly justified. For example, lines 43-44 mention "First, a brain network is a correlation matrix defined on a complete graph." This is not correct as correlation matric computed using methods like Pearson’s Correlation Cofficient is a way to measure brain connectivity. There can be many other ways to represent brain network e.g. Effective Connectivity, Directed connectivity, Entropy based, causality based (Granger causality), etc. Lines 49-51:  it would be better to explain how a 1D time-series of a voxel has positional information.
>
> A: Thanks for pointing this out. We admit this is overclaimed, and we have revised it to “First, one of the simplest and most commonly used methods to construct a brain network in the neuroimaging community is via pairwise correlations between BOLD time courses from two ROIs".
> For the second question, we in fact do not conclude “1D time-series of a voxel has positional information”. Instead, what we mean here is that the relative position of each node (brain region) is encoded when the connection profile is taken as the node feature. Such relative position is discussed in GNN papers such as [1][2], and it should not be confused with the physical positions of brain regions. We will add clarifications about this in the revision.
>
> [1] Li, Pan, et al. "Distance encoding: Design provably more powerful neural networks for graph representation learning." NeurIPS (2020).
> [2] Cui, Hejie, et al. "On positional and structural node features for graph neural networks on non-attributed graphs." KDD-DLG (2021).
>
> > Q2: The authors do not mention how they compute the X∈RV∗V matrix. (How is the matrix X computed?)
>
> A: Thank you for the suggestion. In this paper, since we aim to propose a transformer that is generic and applicable to different brain networks, the most commonly way to construct a brain network is adopted, which we excluded due to the page limit. Take functional brain network as an example, to generate the matrix $X$, a brain atlas or a set of Regions of Interest (ROI) are first selected to define the nodes. Then, the representative fMRI BOLD series from each node are obtained by either averaging or performing SVD on the time series from all the voxels within the node. Various measures have been proposed for assessing brain connectivity between pairs of nodes. We adopted the simplest and most frequently used method in the neuroimaging community where connections are calculated as the pairwise correlations between BOLD time courses from two ROIs. After selecting the Functional Connectivity (FC) measure, the strength of connectivity between each pair of nodes in the brain network is evaluated. Fisher’s transformation is further performed to transform the original FC measures and improve their distribution properties. The transformed FC measures can then be utilized for the subsequent analysis of functional brain networks. We will include these details in the Appendix in the revision.
>
> > Q3: The authors cite multiple self-attention and graph based methods but do not compare with them, for example BrainGNN, STAGIN. It would be better to compare with these methods and others like these on the same task.
>
> A: Thanks for your suggestions. We did not explicitly compare with some of these methods because they have already been compared with FBNetGen and BrainGB (two of the baselines in this work). To provide a direct comparison, we have now explicitly added BrainGNN as one of the baselines in our comparison. For STAGIN, we cited it because it also analyzes brain graphs with transformer models. However, it is designed for dynamic brain graph while we focus on static brain graphs so it is not comparable here.
>
> > Q4: There are others self-attention and graph based methods developed for neuroimaging datasets.
> A Deep Learning Model for Data-Driven Discovery of Functional Connectivity https://doi.org/10.3390/a14030075
> Deep Dynamic Effective Connectivity Estimation from Multivariate Time Series https://arxiv.org/abs/2202.02393
>
> A: Thanks for pointing this out. We agree that the ideas of these two papers are very familiar with FbNetGen. Following your suggestion, we have added the first one into our baselines, since these two papers come from the same authors and they are basically similar models. The result has been updated as ‘BrainnetGNN’ in the updated Table 1 in our first comment for all reviewers above.

---

> ### Author Response · Authors · 2022-08-08
> **Thanks for the initial review; Looking forward to any feedback**
>
> Dear reviewer CYN4,
>
> We are grateful for the opportunities provided in your initial review to improve our work. However, we cannot agree with your conclusion that our work is "with technical flaws, weak evaluation, inadequate reproducibility or so on". In summary, it is very unfair to say the strength of this work is merely 1% improvement in classification accuracy (even if experimental result is the only thing you care about when you read a research paper, our performance improvement is more than 1% if you look at AUROC, which is a more discriminative and comprehensive metric than accuracy because accuracy only considers a specific untuned threshold of 0.5, and beyond that we have clearly shown an ability to effectively learn the latent functional modules underlying brain regions without specific supervision; let alone our problem-level, technical and theoretical analysis). We have provided a comprehensive rebuttal to address every one of your concerns, and hope you could take a look and let us know any of your further concerns.
>
> Best,
> Authors

---

### Official Review · Reviewer_baE1 · 2022-07-11

**Rating:** 6
**Confidence:** 2
**Soundness:** 3 good
**Presentation:** 3 good
**Contribution:** 3 good

**Summary:**

This paper proposed brain network transformer that learns representation from brain network with a transformer-based design. More specifically, the two main designs of the proposed brain network transformer are the multi-head self-attention module and the orthonormal clustering readout. In experiments, the proposed method showed performance improvements over multiple baseline methods.

**Questions:**

Please address the above W1 and W2 if possible.

**Strengths And Weaknesses:**

S1. This paper is overall clearly written and easy to follow.

S2. Brain network analysis is an important field of study and the proposed method is interesting and reasonably designed.

S3. Both the experimental results in the tables and the case study features in Section 4.4 validates the effectiveness of the proposed method.

W1. In Introduction, the authors mentioned the advantage on efficiency of the proposed method over existing graph transformer models. Hence, runtime experiments would be helpful to validate this claim.

W2. Other than the transformer-base GNN methods, I wonder how does normal GNN perform on the brain networks. I would suggest the authors to also include vanilla GNN baselines for comparison.

---

> ### Author Response · Authors · 2022-08-02
> **Authors' initial rebuttal to Reviewer baE1**
>
> We thank the reviewer for the time, valuable feedback, and suggestions for improvements. For your several questions, we clarify as follows.
>
> > Q1. In Introduction, the authors mentioned the advantage on efficiency of the proposed method over existing graph transformer models. Hence, runtime experiments would be helpful to validate this claim.
>
> A: We added the runtime experiments as shown in the following table. We will include these results and discussions in the revised paper.
>
> | Dataset |   Method   | Running Time (min) |
> |:-------:|:----------:|:------------------:|
> |  ABIDE  | BrainNetTF |     2.45±0.04      |
> |  ABIDE  | VanillaTF  |     2.32±0.10      |
> |  ABIDE  |    SAN     |     93.01±0.96     |
> |  ABIDE  | Graphormer |    133.52±0.54     |
> |  ABCD   | BrainNetTF |     41.31±1.16     |
> |  ABCD   | VanillaTF  |     36.26±2.12     |
> |  ABCD   |    SAN     |     908.0±3.6      |
> |  ABCD   | Graphormer |    4089.86±5.7     |
>
> As we can observe, state-of-the-art models of Graphormer and SAN are much slower than our BrainNetTF and VanillaTF. This is mainly because their implementations are not optimized towards the unique properties of brain networks. Specifically, let $e$ be the number of edges and $v$ be the number of nodes. The computations of Graphormer and SAN are optimized for the case where $e \ll v^2$. However, brain networks usually have a small number of nodes but dense connections, i.e., $e \simeq v^2$. Therefore the optimized sparse matrix operations in Graphormoer and SAN do not work as intended. On the other hand, since the number of nodes in brain networks is usually quite small (less than 500), we can directly speed up the calculation using matrix multiplication, which is what we did in BrainNetTF and VanillaTF. Besides, the edge feature generation operator in Graphormer further increases the burden on its computing time.
>
> > Q2. Other than the transformer-base GNN methods, I wonder how does normal GNN perform on the brain networks. I would suggest the authors to also include vanilla GNN baselines for comparison.
>
> A: In fact, we included results of a vanilla GNN-based baseline, BrainGB, in our original draft, which represents the best-performed vanilla GNN taken from a recent benchmark paper of GNNs on brain networks. This baseline has been shown to be consistently stronger than vanilla GCN, GAT, etc. on multiple brain network datasets including ABCD. Besides, based on other reviewers' suggestions, we have added BrainGNN [1], BrainNetGNN [2] and DGM [3] as additional GNN-based baselines, with Sensitivity and Specificity as additional evaluation metrics. As can be observed from the updated Table 1 in our first comment for all reviewers above, all of BrainGNN, BrainNetGNN, and DGM perform much worse than BrainNetTF on both ABCD and ABIDE.
>
> [1] Li, Xiaoxiao, et al. "Braingnn: Interpretable brain graph neural network for fmri analysis." Medical Image Analysis 74 (2021): 102233.
> [2] Mahmood, Usman, et al. "A deep learning model for data-driven discovery of functional connectivity." Algorithms 14.3 (2021): 75.
> [3] Kazi, Anees, et al. "Differentiable graph module (dgm) for graph convolutional networks." IEEE Transactions on Pattern Analysis and Machine Intelligence (2022).

---

> > ### Comment · Reviewer_xfsP · 2022-08-03
> > **Quick question**
> >
> > Just to be sure we are all on the same page, what is included in "running time"? I'm guessing training, in which the variation corresponds to the five times the process is repeated with different seeds? Or could it be the evaluation across the entire dataset? Usually in performance evaluations there's an important distinction between training and inference time, which I think is important to clear out here.

---

> > > ### Author Response · Authors · 2022-08-03
> > > **Answer to the quick question on "running time"**
> > >
> > > Thanks for asking! Yes, "running time" here is referring to the training time, and we got the results (mean and std in the table) after five repeated runs with different seeds. The inference times are much shorter and not that different across the compared algorithms so we did not include them. We will clarify this with an updated proper table name in the revised paper.

---

> ### Author Response · Authors · 2022-08-08
> **Thanks for the initial review; Looking forward to any feedback**
>
> Dear reviewer baE1,
>
> Many thanks for the opportunities to improve our work. We have addressed your W1 and W2 in the rebuttal and are looking forward to any feedback and further questions you might have.
>
> Best,
> Authors

---

> > ### Comment · Reviewer_baE1 · 2022-08-08
> > **response to authors**
> >
> > I appreciate the authors' detailed response as well as the discussion with the other reviewers. I think the authors have addressed my concerns and I've updated my score.

---

> > > ### Author Response · Authors · 2022-08-08
> > > **Thank you**
> > >
> > > Dear reviewer baE1,
> > >
> > > Many thanks for your positive feedback.
> > >
> > > Best,
> > > Authors

---

### Author Response · Authors · 2022-08-02
**(For all reviewers) updated main result table with added baselines and metrics**

Table 1: updated main result table with added baselines and metrics.

| Dataset |     Method     |    AUC    |    ACC    | Sensitivity | Specificity |
|:-------:|:--------------:|:---------:|:---------:|:-----------:|:-----------:|
|  ABIDE  |      SAN       | 71.3±2.1  | 65.3±2.9  |  55.4±9.2   |  68.3±7.5   |
|  ABIDE  |   Graphormer   | 63.5±3.7  | 60.8±2.7  |  78.7±22.3  |  36.7±23.5  |
|  ABIDE  |   VanillaTF    | 76.4±1.2  | 65.2±1.2  |  66.4±11.4  |  71.1±12.0  |
|  ABIDE  |    BrainGB     | 69.7±3.3  | 63.6±1.9  |  63.7±8.3   |  60.4±10.1  |
|  ABIDE  |  BrainnetCNN   | 77.4±2.4  | 70.4±2.7  |  64.8±9.7   |  72.3±10.2  |
|  ABIDE  |  BrainGNN [1]   | 62.4±3.5  | 59.4±2.3  |  36.7±24.0  |  70.7±19.3  |
|  ABIDE  |    FBNETGNN    | 77.6±1.2  | 70.0±1.4  |  66.7±8.7   |  64.4±9.2   |
|  ABIDE  | BrainNetGNN [2] | 55.3±1.9  | 51.2±5.4  |  67.7±37.5  |  33.9±34.2  |
|  ABIDE  |     DGM [3]     | 52.7±3.8  | 60.7±12.6 |  53.8±41.2  |  51.1±40.9  |
|  ABIDE  |   BrainNetTF   | 80.2±1.0  | 71.0±1.2  |  68.4±7.5   |  74.8±7.9   |
|:-------:|:--------------:|:---------:|:---------:|:-----------:|:-----------:|
|  ABCD   |      SAN       | 90.1±1.2  | 81.0±1.3  |  84.9±3.5   |  77.5±4.1   |
|  ABCD   |   Graphormer   | 89.0±1.4  | 80.2±1.3  |  81.8±11.6  |  82.4±7.4   |
|  ABCD   |   VanillaTF    | 94.3±0.7  | 85.9±1.4  |  87.7±2.4   |  82.6±3.9   |
|  ABCD   |    BrainGB     | 91.9±0.3  | 83.1±0.5  |  84.6±4.3   |  81.5±3.9   |
|  ABCD   |  BrainnetCNN   | 93.5±0.3  | 85.7±0.8  |  87.9±3.4   |  83.0±4.4   |
|  ABCD   |  BrainGNN [1]   |    OOM    |    OOM    |     OOM     |     OOM     |
|  ABCD   |    FBNETGNN    | 94.5±0.7  | 87.2±1.2  |  87.0±2.5   |  86.7±2.8   |
|  ABCD   | BrainNetGNN [2] | 75.3±5.2  | 67.5±4.7  |  67.7±5.7   |  68.0±6.5   |
|  ABCD   |     DGM [3]     | 76.8±19.0 | 68.6±8.1  |  40.5±29.7  |  95.6±4.2   |
|  ABCD   |   BrainNetTF   | 96.2±0.3  | 88.4±0.4  |  89.4±2.6   |  88.4±1.5   |

[1] Li, Xiaoxiao, et al. "Braingnn: Interpretable brain graph neural network for fmri analysis." Medical Image Analysis 74 (2021): 102233.
[2] Mahmood, Usman, et al. "A deep learning model for data-driven discovery of functional connectivity." Algorithms 14.3 (2021): 75.
[3] Kazi, Anees, et al. "Differentiable graph module (dgm) for graph convolutional networks." IEEE Transactions on Pattern Analysis and Machine Intelligence (2022).

---

### Author Response · Authors · 2022-08-06
**(For all reviews) thanks again for the reviews and we look forward to hearing your feedback on our rebuttal**

Dear reviewers,

We are grateful for your initial reviews to help clarify and improve our work.

We thank reviewer xfsP for the further active feedback on our rebuttal and wonder what is the current score. In the meantime, we very much look forward to hearing the feedback from other reviewers as well.

Best, Authors

---

### Meta-Review · Area_Chair_uJ7Y · 2022-08-26

**Recommendation:** Accept
**Confidence:** Less certain

**Metareview:**

This paper was reviewed by four reviewers. Three reviewers have participated in the discussions and they are all finally convinced and provided positive recommendations. The fourth reviewer was negative about this paper with some concerns. The authors provided very detailed rebuttals, but the reviewer did not respond to rebuttals though repeated reminders. I checked the comments and rebuttals and tend to believe most of the concerns of this reviewer have been addressed, at least to a large extent. Thus I recommend this paper to be accepted at this point.

**Award:**

No

---

### Decision · Program_Chairs · 2022-09-14

Accept